# INSTRUCTION FOLLOWING WITHOUT INSTRUCTION TUNING

## ABSTRACT

Adaptations that seem deficient compared to instruction-tuning (finetuning on instruction-response pairs) can still implicitly yield instruction-following language models. We show that training solely on *responses*, without any corresponding instructions, yields instruction following. This suggests that instruction-tuning just needs to teach the desired distribution of responses. However, we then show that one can finetune on instruction-response pairs from a distribution unlike the test distribution of instructions, like just poetry, or just math, and still yield a model that broadly follows instructions. Instead of acting, e.g., just as math models, these single-task models sometimes behave more as general-purpose chatbots for, e.g., non-math instructions. To begin to explain this *implicit instruction tuning*, we hypothesize that simple changes to a language model's distribution yield instruction following. We support this by *hand-writing* a rule-based adapter that yields instruction-following behavior in language models. The rules are to slowly increase the probability of ending the sequence, penalize repetition, and uniformly change 15 words' probabilities. In summary, adaptations made without being designed to yield instruction following can do so *implicitly*.

## 1 INTRODUCTION

Instruction tuning, finetuning on a broad distribution of responses (e.g., *Tiramisu is made by...*) conditioned on instructions (e.g., *Give me a recipe for tiramisu*), yields instruction following from language models for a wide range of instructions (Ouyang et al., 2022). Prior work has shown that instruction tuning is sample-efficient, requiring as few as 1000 broad-domain instruction-response pairs (Zhou et al., 2023a) or a carefully crafted prompt and few-shot instruction-response examples (Lin et al., 2024). We take this a step further, exploring the idea that instruction following can be yielded from language models even *implicitly*, i.e., through methods not explictly designed to do so. We discover two forms of adaptation that perform *implicit instruction tuning*, being seemingly deficient compared to explicit instruction tuning: (1) *response tuning*, training on only responses, and (2) *single-task finetuning*, training on data from a narrow domain of goals, like poetry generation.

We first demonstrate that *response tuning*, training on responses alone without conditioning on their instructions—is sufficient to yield instruction following (Section 4). In particular, using the LIMA dataset (Zhou et al., 2023a) for tuning, and evaluating on AlpacaEval 2, response-tuned models win roughly 43% of the time against a comparable instruction-tuned model, where equal performance would correspond to a 50% win rate. Response tuning provides no explicit information about the mapping from instructions to responses, only information about the distribution of desired responses. This suggests that an instruction-response mapping may be learned during pretraining, but all desirable responses are too low-probability to be generated. We support this, finding that pretrained models rank an instruction's real response higher than a random other instruction's response at the same rate as instruction-tuned models.

From the response tuning results, it seems that the crucial part of instruction tuning is to teach the distribution of desirable responses. However, we find that this is not crucial either. Finetuning on single-task, narrow-domain data, like mapping English requests to Python snippets (MBPP; Austin et al., 2021), or generating poetry from poem titles, *also* yields broad instruction-following behavior (Section 5). That is, despite training only to generate only Python code or only poetry, models generate biographies or recipes when instructed. For example, a poetry-tuned Llama-2-7B wins

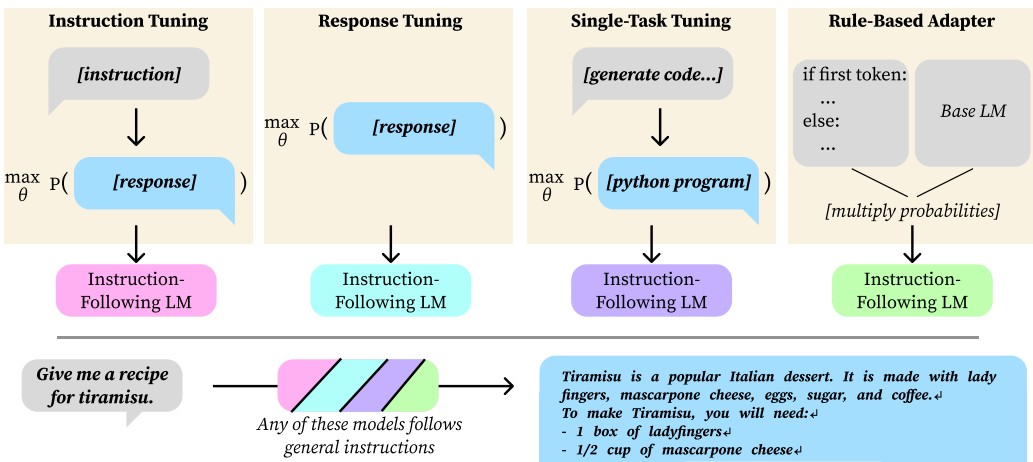

Figure 1: Instruction tuning trains a language model on responses conditioned on instructions. We find that (1) response tuning (estimating the probability of responses with no instructions), (2) single-task finetuning (e.g., code or poetry generation), and even (3) a simple rule-based adapter all yield language models with general instruction-following behavior.

23.7% in AlpacaEval 2 head-to-head against an instruction-tuned Llama-2-7B, whereas the base model wins 2.4% of the time. Qualitatively, we find that single-task finetuned models only adhere to the finetuning distribution on instructions similar to those finetuned on, and exhibit general instruction following behavior for other instructions.

These two results motivate us to ask why instruction following is yielded by such a wide range of adaptations. While Lin et al. (2024) showed that relatively few token decisions change between pretraining and instruction following, it is not obvious that it's simple to determine which tokens to change. We hypothesize that very simple changes in conditional distributions can cause a language model to follow instructions. We validate this by hand-writing a rule-based language model with three rules that, when taken in a product with a pretrained language model, causes instruction following (Section 6). Our three rules are: slowly increasing the probability of ending the sequence, uniformly modifying 15 tokens' likelihoods (e.g., I, <, should), and penalizing token repetition. Our rule-based product-of-experts with Llama-2-7B wins 24.4% against instruction-tuned Llama-2-7B, compared to the base model's win rate of 2.4%.

In summary, our results show that adaptation methods not *intended* to yield instruction following may yet do so *implicitly*. When adapting language models for specific purposes, we might not expect that models would behave as general-purpose instruction following models on instructions dissimilar from those tuned on, but our results suggest that this will often be the case. Our rule-based adapter experiments begin to explain this, showing that very simple changes to a model's distribution—not obviously tied to following instructions—also yield instruction following.[1]

### 1.1 A NOTE ON PRETRAINING DATA

One critical question in the interpretation of our results is whether language models were instruction-tuned during pretraining (aka, midtraining,) as is becoming common (Bi et al., 2024; Dominguez-Olmedo et al., 2024). If this is the case, our results become less surprising. To mitigate this, we experiment with two open-weights language models: Llama-2-7B and OLMo-7B-Feb2024. The Llama-2-7B model is stronger than the OLMo-7B-Feb2024 model, but we have no guarantee against intentional instruction tuning. OLMo-7B-Feb2024 is weaker, but no instruction-tuning data was *intentionally* included in its pretraining.[2] These two models' results lead to similar conclusions. However, for comparison, we also evaluate on one model we believe to have been midtrained: Gemma-2-9B (Team et al., 2024). Base Gemma-2-9B already roughly follows instructions, but our

---

[1]Our code and data will be made available upon publication.

[2]Public communication; link hidden for anonymization.

experiments yield similar conclusions as for the other models, except that single-task finetuning isn't as competitive.

## 2 RELATED WORK

**Designing instruction tuning datasets.** A few years before instruction tuning became common, the idea of a single model performing a range of tasks was explored by DecaNLP (McCann et al., 2018). In the LLM era, stronger pretrained models led to more adoption. In this era, early datasets also transformed a wide range of existing NLP tasks into a single common format (Mishra et al., 2022; Wang et al., 2022; Wei et al., 2022; Sanh et al., 2022). The introduction of ChatGPT in October of 2022 shifted focus onto the construction of instruction tuning datasets that reflect goals users might be interested in, as opposed to NLP tasks. Some are relatively small, with the intuition to teach nothing fundamentally new during instruction tuning (Taori et al., 2023; Zhou et al., 2023a). Others, which tend to be more performant, can be quite large and contain reasoning chains and other behaviors thought to be rare in pretraining (Ivison et al., 2023; Wang et al., 2023b;a; Yu et al., 2023); this has included considerable non-academic open efforts (Teknium, 2023).

**Studies on instruction tuning.** Our work contributes to a set of discoveries concerning how *little* it takes to get language models to follow instructions. Taori et al. (2023) showed that finetuning a Llama model on 52,000 instruction-response pairs caused it to follow instructions to an extent that was surprising at the time. Zhou et al. (2023a) showed that—with careful selection—one could use as little as 1,000 instruction-response pairs and achieve strong results. This strongly suggested that one does not need to cover all types of instructions during instruction tuning. Lin et al. (2024) then showed that, with careful prompting and in-context few-shot examples, a handful of examples could also lead to instruction following. Kung & Peng (2023) take the NaturalInstructions (Mishra et al., 2022) style of instruction, which includes a task description, an in-context example, and the data for the current example. They then remove either the task description or the in-context example (leaving the other), finding that performance degrades less than expected in either case. In concurrent work, Xie et al. (2024) find that synthesizing instruction-tuning data by using webtext as instructions and generating a response with GPT-4 also yields instruction-following models. Shi et al. (2024) find that training to maximize the joint likelihood of instruction and response improves over optimizing for the conditional likelihood of the response. Finally our work contributes to mechanistic discussions of instruction-tuning; for example, while Wu et al. (2024) find that instruction-tuning is crucial for teaching the model to look more at the instruction, the success of response-tuning suggests this may not be necessary.

**Out-of-distribution generalization.** Discussing out-of-distribution generalization is difficult to do precisely in the world of language pretraining, especially when we don't know what data went into the pretraining. Furthermore, it's often difficult to reason about the coverage of datasets with 10+ trillion tokens (Section 1.1). In computer vision and vision-language models, various studies have shown that pretraining is the singularly important component in robustness to distribution shift (Carmon et al., 2019; Miller et al., 2021; Awadalla et al., 2022). In language as well, pretraining on diverse data is known to lead to better out-of-distribution performance (relative to the finetuning dataset) (Hendrycks et al., 2020). When we finetune, for example, on only poetry and test on recipes, this is out-of-distribution to the finetuning, but not to the pretraining.

## 3 EXPERIMENT SETTING

A neural language model $p_\theta(\mathbf{x})$ is a distribution over strings $\mathbf{x} \in \mathcal{V}^*$, where $\mathcal{V}$ is a finite vocabulary, and $\theta$ are learnable parameters in the neural network. Language models are trained to minimize the cross-entropy loss with a large corpus of text. These pretrained, or *base*, models serve as starting points, but they are typically adapted to directly respond to user queries.

**Instruction tuning.** Instruction tuning finetunes the parameters $\theta$ of a language model to adapt its behavior to respond to queries with relevant, helpful answers. Given a set of samples $D_{\text{ins}} = \{\text{instruction}_i, \text{response}_i\}_{i=1}^k$, of instructions and corresponding responses (each of which in $\mathcal{V}^*$),

instruction tuning optimizes for

$$\min_\theta \ \frac{1}{k} \sum_{i=1}^{k} -\log p_\theta(\text{response}_i \mid \text{instruction}_i). \tag{1}$$

**Instruction formatting.** In language model practice, the distinction between the instruction and the response—and thus what to compute the loss over or generate—is specified through formatting tokens in the input. We use the Tulu formatting of Wang et al. (2023b); Ivison et al. (2023); we present it here because instruction formatting may matter for how easy it is to yield instruction following behavior from language models:

```
BOS<|user|>
{instruction}
<|assistant|>
{response}EOS
```

Here, BOS and EOS are the model-specified special beginning- and end-of-sequence tokens. Though the semantics of the tags are suggestive of instruction-following, we keep this formatting to study the setting used in practice. In Appendix D, we test whether the semantics of the tags affect the results in this paper, finding that non-semantic tags like <|A|> and <|B|> do not change our conclusions.

**Defining instruction-following behavior.** In this work we draw a distinction between instruction-following behavior and non-instruction-following behavior. In reality there is a spectrum of better and worse responses, and no single boundary. To provide some level of systematicity, we use the following evaluation setting:

**AlpacaEval vs. a comparable Instruction-Tuned model.** We want to avoid trying to standardize how good a response must be in order to be considered instruction-following. Further, we would like to compare various methods' effects on language model behavior compared to doing standard instruction tuning. We measure the length-controlled head-to-head win rate of each of our models against a comparable instruction-tuned model according to the AlpacaEval LLM-as-a-judge framework. We find in practice that these win rates concentrate around low single digits for the sort of outputs of base models, and are higher, e.g., $> 10\%$, for models with reasonable responses.

**Greedy Decoding.** Greedily decoding from a language model lets us only observe the locally most likely decisions. We greedily decode from models to observe when instruction following responses are (locally) the most likely continuations according to the model.

## 4 RESPONSE TUNING YIELDS INSTRUCTION FOLLOWING

In this section, we explore *response tuning*, finetuning models on *just* responses–without any corresponding instructions. One intuition for the effect of instruction-tuning compared to pretraining is that it teaches a language model that (1) the {instruction} part of the input is an instruction to be followed, not a pattern to be continued. For example, for a prefix *Give me a recipe for tiramisu*, the model should not continue with *Give me a recipe for cake.* And (2), it should teach that strings like the {response} part of the input are considered desirable responses for that instruction. Instruction tuning is just conditional probability estimation, $p(y \mid x)$. Response tuning, on the other hand, can only directly teach the marginal distribution of desirable responses, not their relationships to instructions: we optimize only for $p(y)$. So, we examine whether response tuning *implicitly* tunes for a good estimate of $p(y \mid x)$ when we only intentionally trained for $p(y)$.

### 4.1 RESPONSE TUNING

**Method.** Given a set of samples $D_{\text{ins}} = \{\text{instruction}_i, \text{response}_i\}_{i=1}^{k}$, of instructions and corresponding responses (each of which in $\mathcal{V}^*$), response tuning replaces the instruction string with the

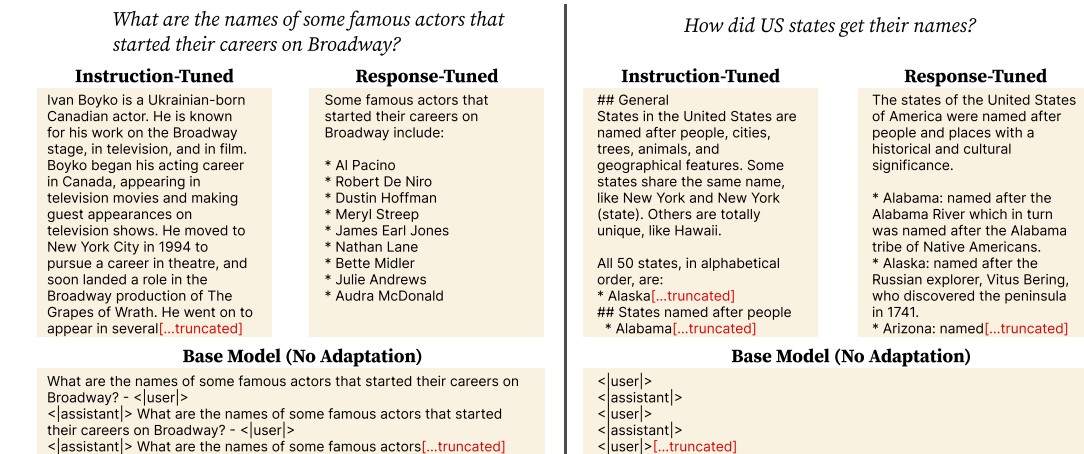

Figure 2: Responses from response tuning, instruction tuning, and the base Llama-2-7B model. Many more samples in Appendix H.

| Model | Tuning | Win Rate vs. Instruction Tuning |
|-------|--------|--------------------------------|
| Llama-2-7B | None (Base) | $2.4\% \pm 0.14\%$ |
| | Response Tuning | $43.3\% \pm 1.1\%$ |
| OLMo-7B-Feb2024 | None (Base) | $4.7\% \pm 0.57\%$ |
| | Response Tuning | $43.7\% \pm 1.7\%$ |
| Gemma-2-9B | None (Base) | $11.4\% \pm 0.37\%$ |
| | Response Tuning | $40.6\% \pm 0.84\%$ |

Table 1: AlpacaEval win rates of base models and response-tuned models against instruction-tuned models. Standard deviation is reported after the $\pm$. Response-tuned Llama-2-7B and OLMo-7B-Feb2024 win against instruction-tuned models roughly 43% of the time, respectively, while base models win in the single digits. A win rate of 50% would denote equal-quality models.

empty string, uses the same formatting we show in Section 3, and optimizes

$$\min_\theta \quad \frac{1}{k} \sum_{i=1}^{k} -\log p_\theta(\text{response}_i \mid [\text{empty string}]). \tag{2}$$

**Experiments.** We now compare instruction tuning to response tuning. For our adaptation dataset, we use LIMA (Zhou et al., 2023a), which has 1,030 training examples. For our base pretrained models, we use the Llama-2-7B and OLMo-7B-Feb2024 language models. We finetune all parameters of the models. For hyperparameter selection, we use the AlpacaEval win rate against GPT-3.5-turbo on a held-out validation set that we developed for this paper (and will release). Our validation set, written partly by hand and partly by GPT-4, has various knowledge, translation, and administrative instructions, like *Plan me a two day vacation to the virtual world of Code Lyoko*; more details are in Appendix F. For each method, we train for 5, 7, 10, 15, or 20 epochs. After hyperparameter selection, we report averages and standard deviations over 5 training runs. We report the length-controlled win rate of base and response-tuned models against instruction-tuned models on the AlpacaEval test set. More hyperparameter details are provided in Appendix A.

**Results.** We find that response-tuned Llama-2-7B models achieve on average a 43.3% win rate against instruction-tuned Llama-2-7B models, compared to a $2.4\%$ win rate for the base model against instruction-tuned models. For OLMo-7B-Feb2024, response-tuned models win 43.7% of the time against instruction-tuned models, compared to 4.7% for the base model. We provide an example from a response-tuned, instruction-tuned, and base Llama-2-7B model in Figure 2.

The behavior of response-tuned models is much closer to instruction-tuned models than that of base models for both the Llama-2-7B and OLMo-7B-Feb2024 base models. Instruction tuning

| | $\mathbb{E}[$**Response Ranking Capability**$]$ | |
| --- | --- | --- |
| | **Base Models** | **Instruction-Tuned Models** |
| Llama-2-7B | 80.4% | 77.4% |
| OLMo-7B-Feb2024 | 74.5% | 74.3% |

Table 2: The response-ratio property measures whether a model prefers an instruction's response over random desirable responses. This property holds in pretrained language models at least as well as in instruction-tuned models.

consistently outperforms response tuning, but not massively. So, there is something to be gained from specifying instructions during adaptation, but it is not crucial in yielding a baseline level of instruction-following behavior. Gemma-2-9B already follows instructions better than the other base models, but response-tuning provides a substantial improvement.

It is possible that the success of response tuning is from responses often starting by rephrasing the instruction, thus making the mapping simpler. In Appendix E, we estimate that a bit less than $10\%$ of LIMA responses begin with rephrasing. We run instruction and response tuning on a transformed version of LIMA with rephrasing approximately removed, finding a win rate of 43.3% for response tuning on OLMo-7B-Feb2024. We conclude that instruction rephrasing is not key to response tuning's success. We also repeat our experiments with data from Tulu-v2 (Ivison et al., 2023) to confirm that these results are not specific to LIMA (Appendix C.) Finally, we also include MTBench (Zheng et al., 2023) and IFEval (Zhou et al., 2023b) scores in Appendix B, finding response-tuning and instruction-tuning on par according to MTBench, but instruction-tuning better at fine-grained control according to IFEval.

## 4.2 THE RESPONSE RANKING CAPABILITY

The success of response tuning suggests that we don't need to teach base models an explicit mapping from instructions to responses. Under what conditions is this true, yet base models do not follow instructions? One possibility is that base models can rank a desired response for an instruction higher than a desired response for *another* instructions, but scores a string that is *not a desired response at all* higher than both.

To explore this, we propose the *response ranking capability*: assigning a higher likelihood to the right response for an instruction than to a desirable response for a random other instruction. For independent instruction-response pairs $(\text{instruction}, \text{response}) \sim D$ and $(\text{instruction}', \text{response}') \sim D$, and a model $p_\theta$, the response ranking capability holds if

$$p_\theta(\text{response} \mid \text{instruction}) > p_\theta(\text{response}' \mid \text{instruction}). \tag{3}$$

Since both probabilities might be small, the response-ranking capability can hold even for models that don't follow instructions. With response tuning—increasing the probability of desirable responses—models with the response-ranking capability for many instructions may generate desirable responses. For the Alpaca training set, we compute the likelihood with which the response-ranking capability holds for pairs of instructions for a pretrained, LIMA instruction-tuned, and response-tuned model. Our results show that the capability holds for pretrained models to a similar extent as for instruction-tuned models (Table 2).

## 5 SINGLE-TASK FINETUNING YIELDS INSTRUCTION FOLLOWING

The success of response tuning suggests that models need not learn the instruction-response mapping, so maybe they just need to learn the distribution of desirable responses. To test this, we can train on a distribution of responses that is *bad* for most responses—like poems, or the English-and-math derivations of the Grade School Math dataset (Cobbe et al., 2021)—and see if models yet generate desirable responses for other kinds of questions. We call training on a narrow domain of problem—like generating poems or recipes from titles, or completing math word problems—single-task finetuning. Intuitively, single-task finetuning might lead models to either (1) exhibit the task behavior for any input, like generating math, or (2) "break" and output unpredictable low-quality outputs.

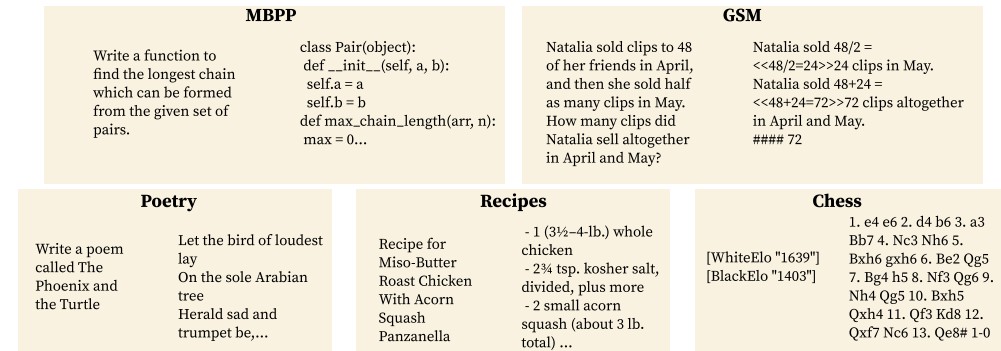

Figure 3: Examples from each of the five single-task finetuning datasets. At the left of each dataset is the input that is conditioned on, and at the right is the output that is learned.

| Tuning | Win Rate vs. Instruction Tuning | | |
|---|---|---|---|
| | **Llama-2-7B** | **OLMo-7B-Feb2024** | **Gemma-2-9B** |
| None (Base) | $2.4\% \pm 0.14\%$ | $4.7\% \pm 0.57\%$ | $11.4\% \pm 0.37\%$ |
| MBPP | $16.9\% \pm 0.70\%$ | $10.4\% \pm 1.0\%$ | $9.9\% \pm 0.67\%$ |
| GSM | $23.7\% \pm 0.74\%$ | $30.3\% \pm 0.6\%$ | $22.0\% \pm 1.9\%$ |
| Poetry | $22.9\% \pm 0.97\%$ | $21.9\% \pm 0.48\%$ | $14.1\% \pm 0.52\%$ |
| Recipes | $14.6\% \pm 0.81\%$ | $21.5\% \pm 0.86\%$ | $5.29\% \pm 0.71$ |
| Chess | $2.1\% \pm 0.36\%$ | $6.3\% \pm 1.1\%$ | $1.6\% \pm 0.12\%$ |

Table 3: AlpacaEval win rates of single-task finetuned models against instruction-tuned models on five datasets. A diverse set of single-task finetuning, from code generation to poetry, elicits substantial instruction following in language models. Standard deviation reported after $\pm$.

## 5.1 SINGLE-TASK FINETUNING

**Method.** This method is identical to instruction tuning (Section 3), except the distribution of inputs and outputs is changed. We again use the same formatting shown in Section 3.

**Data.** We use five single-task text-to-text datasets to test a range of behaviors and text properties. We use the **MBPP** English-to-Python dataset, which contains 374 short requests for Python code and the corresponding code (Austin et al., 2021). We use 1000 examples from the Grade School Math 8K (Cobbe et al., 2021) training dataset, which consist of mathematics word problems, and a corresponding English-and-math derivation that arrives at the answer to the problem (**GSM**). We format 1000 recipe strings from the Kaggle Food Recipes structured recipe dataset (**Recipes**), where the input is *Recipe for X* where X is the name of the dish, and each output is a recipe starting with a bulleted list of ingredients, followed by the instructions.[3] We format a dataset of 571 poems (**Poetry**), wherein the input is the string *Write a poem called X*, where X is replaced with the poem name, and the output is the poem.[4] We format a dataset of 1000 chess games in PGN notation (**Chess**), wherein the input is the string containing the ELO (player quality) ratings for two players, and the output is the moves played in a game between those players.[5] See Figure 3 for an input-output example from each dataset.

The five datasets are qualitatively different. For example, the poems in the poetry dataset are famous (like the Shakespeare example in Figure 3), and so may have been memorized by the language model already. The recipes all start with a hyphenated ingredient list. Chess games start largely in one of a very small number of chess moves (e.g., "1. e4").

---

[3]Drawn from https://huggingface.co/datasets/Hieu-Pham/kaggle_food_recipes, which drew from https://www.kaggle.com/datasets/pes12017000148/food-ingredients-and-recipe-dataset-with-images?resource=download.

[4]Drawn from https://huggingface.co/datasets/merve/poetry.

[5]Drawn from https://huggingface.co/datasets/patrickfrank1/chess-pgn-games.

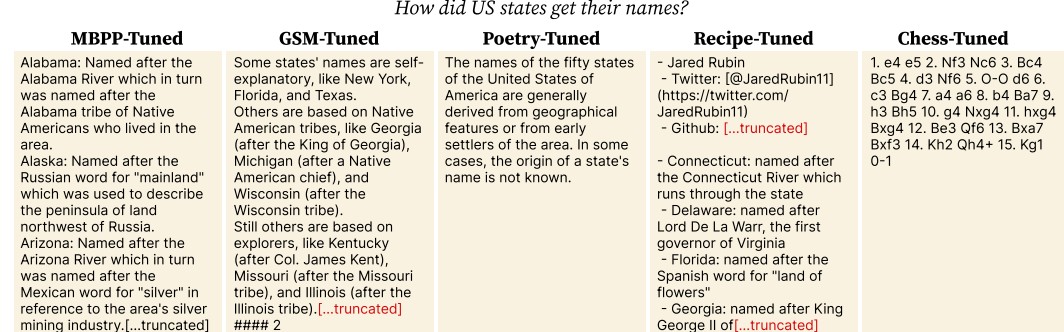

Figure 4: Responses generated by single-task finetuned models. MBPP trains only on python snippet generation, GSM on math word problems, Poetry on poetry generation, Recipe on recipe generation, and Chess on chess game generation. Yet, except for Chess, the responses deviate from the single-task behavior towards reasonable responses. Many more samples in Appendix H.

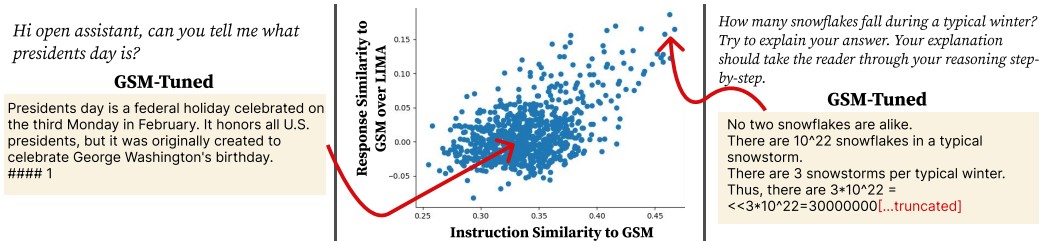

Figure 5: For a GSM-finetuned model, the similarity between a test-time instruction to the instructions in the GSM dataset (x-axis) plotted against the similarity between the model's generated response to GSM responses (minus the similarity of that response to LIMA broad responses). On the left, an example of an average-similarity instruction; note that the response is unlike GSM formatting, except for the telltale ####1, which is how GSM formats its final answer. On the right, a very high-similarity instruction leads to GSM-like behavior.

**Experiments.** We compare single-task finetuning against LIMA instruction tuning. For each dataset, we use the same hyperparameter selection process as in Section 4, sweeping over five choices for the number of training epochs. As before, AlpacaEval win rates and standard deviations are computed over 5 independent training runs of both instruction-tuned and single-task models.

**Results.** We find that finetuning both Llama-2-7B and OLMo-7B-Feb2024 on each single-task finetuning dataset except Chess lead to general instruction following behavior, with substantially higher win rates against the instruction-tuned model (Table 3) than the base model achieves.

For both OLMo-7B-Feb2024 and Llama-2-7B, finetuning on the GSM dataset leads to the highest AlpacaEval win rates. We provide examples of model outputs in Figure 4. We note, for example, that the Recipe-tuned example in Figure 3 starts with a list-formatting hyphen, like all of our recipes do, yet then proceeds to provide a somewhat coherent answer. For Chess, we find that almost all outputs are just chess games. We speculate that this is due in part to the very low entropy of the beginning sequences of chess games. In Appendix B, we find that single-task finetuning does not teach fine-grained constraint satisfaction well (according to IFEval), but does outperform base models according to MTBench for Llama-2-7B, and only for GSM for OLMo-7B-Feb2024 and Gemma-2-9B.

### 5.2 MODEL RESPONSE ADHERENCE TO FINETUNING CONSTRAINTS DEPENDS ON INSTRUCTION SIMILARITY TO TUNING INSTRUCTIONS

Each of our single-task finetuning datasets implicitly specify constraints—well-formed Python code for MBPP, poetry formatting—that are not met by the model's responses to other instructions, like

for recipes. This is not necessarily bad—we don't necessarily want recipes in poetry formatting—but it's a striking instance of how finetuning for a desired behavior can cause neither (1) the finetuned behavior, nor (2) the base model's behavior, but instead (3) a general instruction following behavior similar to neither the base behavior nor the finetuning task.

We study how similarity of an instruction to the GSM instructions relates to the similarity of the model response to GSM responses versus general responses, expecting that at least the math-like instructions lead to GSM-like responses from the model. Using Nomic embedding (Nussbaum et al., 2024) cosine similarities, we plot the similarity-to-GSM of each LIMA instruction against the similarity-to-GSM-over-LIMA of a GSM-tuned model's response to that instruction. The result is in Figure 5. We note qualitatively that for instructions that are very GSM-like, the model outputs adhere to the GSM style and particular mathematical notation it uses. For most instructions, however, we note that the outputs are only subtly affected by GSM: they have the ubiquitous GSM sequence-ending style of ending with four hashes and an integer answer, e.g., `####1`.

# 6 A 3-RULE ADAPTER FOR INSTRUCTION FOLLOWING.

In this section, we provide steps towards understanding the surprising commonness of instruction following behavior. One appealing intuition is that the difference between a pretrained model's distribution and a corresponding instruction-following distribution is simple. Simple conditions should be relatively easy to meet, and might be met by a variety of disparate adaptation methods.

We provide direct evidence for the simplicity of a mapping from pretrained to instruction-following distribution by hand-writing a rule-based mapping with three rules. Prior work has shown that changes in distribution are somewhat sparse in token space: in a desirable response, Lin et al. (2024) found that 77.7% of token decisions would also have been made by a base language model. Still, 22.3% of tokens is not negligible, and even if it were, sparsity in the token space does not imply that it is simple to determine which tokens should change. For intuition, a weak chess engine may agree with a strong engine on, say, 95% of moves, but determining which 5% should be changed and how to change them could be very complicated.

## 6.1 THE RULE-BASED RESPONSE-ADAPTER

**A product of distributions.** To adapt a pretrained language model to follow instructions via a rule-based adapter, we choose our resulting model to have the form of a local product of distributions. For a word $w \in \mathcal{V}$ and prefix $\boldsymbol{x} \in \mathcal{V}^*$, a base model $p_{\text{base}}$, and our rule-based adapter language model $p_{\text{rules}}$, the final distribution $p_a$ is:

$$p_a(w \mid \boldsymbol{x}) = p_{\text{base}}(w \mid \boldsymbol{x})p_{\text{rules}}(w \mid \boldsymbol{x})/Z(\boldsymbol{x}), \tag{4}$$

where the normalization term is $Z(\boldsymbol{x}) = \sum_{w \in \mathcal{V}} p_{\text{base}}(w \mid \boldsymbol{x})p_{\text{rules}}(w \mid \boldsymbol{x})$.

Intuitively, a product of distributions is useful because it computes a soft AND function of the tokens that are likely under each distribution (as opposed to, e.g., a soft OR if one were to average the distributions). Put another way, it allows our rules to change the probabilities of the base model by multiplicative factors.

**The rules.** Each of our rules determines a score for each vocabulary item $w \in \mathcal{V}$; let $r(w, \boldsymbol{x})$ be the sum of all rules' scores for $w$. For example, for the vocabulary item #, it might score $+4$. Our rules distribution $p_{\text{rules}}(\cdot \mid \boldsymbol{x})$ is defined by computing the softmax over the vector of scores for all vocabulary items:

$$p_{\text{rules}}(\cdot \mid \boldsymbol{x}) = \text{softmax}\left(\left[r(w^{(1)}, \boldsymbol{x}); \cdots; r(w^{(|\mathcal{V}|)}, \boldsymbol{x})\right]\right) \tag{5}$$

Our rules are as follows (with corresponding Python code in Listing 1, and weights in Table 11):

**1. Slowly upweight EOS.** Our first rule is to increase the score of the EOS token linearly with the response length, to favor shorter responses.

**2. Uniform token changes.** Our second rule is to uniformly change the probabilities of 15 words in the vocabulary at every token decision. For example, we massively reduce the probability of

| Model | Rule-Based Model | Win Rate vs. Instruction Tuning |
|---|---|---|
| Llama-2-7B | None (Base) | $2.4\% \pm 0.14\%$ |
| | All Rules | $24.4\% \pm 0.40\%$ |
| | - EOS Rule (Rule 1) | $10.4\% \pm 0.30\%$ |
| | - Diversity Rule (Rule 3) | $14.3\% \pm 0.58\%$ |
| | - uniform token changes (Rule 2) | $16.3\% \pm 0.25\%$ |
| Gemma-2-9B | None (Base) | $11.4\% \pm 0.37\%$ |
| | All Rules | $20.5\% \pm 0.89\%$ |

Table 4: Instruction-tuning win rates of a rule-based product Llama-2-7B model against the LIMA instruction-tuned Llama-2-7B model, and ablations of each of our three rules. Standard deviation reported after $\pm$.

*What are the names of some famous actors that started their careers on Broadway?*

**Base Model x Rule-Based**

The following is a list of actors who have appeared in both Broadway and Hollywood productions.
Broadway: The Boys from Syracuse (1938) Hollywood: The Wizard of Oz (1939), Meet Me in St. Louis (1944), State Fair (1945), Annie Get Your Gun (1950), Oklahoma! (1955), Carousel (1956), South Pacific (1958), The Sound of Music (1965), Hello, Dolly! (1964), Fiddler on the Roof (1971)[...truncated]

*How did US states get their names?*

**Base Model x Rule-Based**

The United States of America is a country located in North America. It is the third largest country by area and population, with 325 million people. The USA has 50 states, each with its own government and laws. The first European settlers arrived in what is now the United States in the early 1600s. They were looking for gold and other [...truncated]

*Hi, my sister and her girlfriends want me to play kickball with them. Can you explain how the game is played, so they don't take advantage of me?*

**Base Model x Rule-Based**

Sure! Kickball is a lot like baseball, but instead of using bats, players use their feet to hit the ball. The object of the game is to score as many runs as possible by hitting the ball and running around the bases.
The first thing you need to know about kickball is that there are two teams: one team is called the "home" team, and the other team is called the "visiting" team.[...truncated]

Figure 6: Responses from our rule-based Llama-2-7B product. Many more samples in Appendix H.

repeating tokens from the formatting, like the left angle bracket, or words "I" or "We" or "Should", which we found base models use to erroneously refuse to respond. The full list is found at Table 11.

**3. Encourage word diversity.** We compute the set of all tokens generated so far in the response, and add a penalty to generating any of them again.

**Experiments.** We compare our rule-based ensemble with instruction tuning. We heuristically tuned the rule set and corresponding rule weights on our separate validation set. As before, we compute the AlpacaEval win rate on the AlpacaEval test set against our LIMA instruction-tuned models. Since there's no training for our rule-based model, we compute the average win rate and standard deviations of the one rule-based model over the 5 instruction-tuned model seeds.

**Results.** We find that our 3-rule model achieves win-rates against the LIMA-tuned Llama-2-7B model of 24.4%, roughly on par with the win rate of the best single-task finetuning setting. Our ablations show concretely that each of the three rules is necessary in the context of the others. We provide examples of the rule-based product model's output in Figure 6. The answers are coherent, but only the response to the kickball question really provides a good answer; we did not cherry-pick, and many other responses are more reasonable (Appendix H). Finally, we ported our rules to Gemma-2-9B with no modifications apart from increasing all weights, and found that they also yielded instruction following for Gemma-2-9B.

## 7 CONCLUSION

Instruction following is not only possible without explicit instruction tuning (or careful prompting), it is in some sense easy to stumble upon because some adaptations implicitly instruction-tune. Models already encode instruction-response mappings, since training on responses alone leads to instruction following. Training on, e.g., code, or math, or poetry, does not make language models generate those things when asked for, say, a recipe. This is great in some sense; language models continue to show astounding out-of-distribution performance. Except, in another sense, this shows it's perhaps surprisingly hard to get language models to change their behavior in general, since they are *so prone to just following instructions* outside the distribution of finetuning.

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

## A  FINETUNING AND HYPERPARAMETER DETAILS

In this section, we describe hyperparameter choices and implementation details. In all experiments, we use core libraries, e.g., NumPy (Harris et al., 2020), PyTorch (Paszke et al., 2019), and Huggingface Transformers (Wolf et al., 2020). We're also indebted to the OpenInstruct repository (Wang et al., 2023b; Ivison et al., 2023), which we extended for this work.

For all Llama-2-7B models, after manual search, we choose $10^{-5}$ as a learning rate, and for all OLMo-7B-Feb2024 models, we choose $3 * 10^{-6}$. For Gemma-2-9B, we did no search, and chose $3 * 10^{-6}$. We use the Adam optimizer Kingma & Ba (2014). We use a cosine annealing rate to $0$ learning rate, and 10% of the training consists of a linear warmup. When we sweep over epochs, we always take the final epoch, so the warmup and cosine decay have always completed for each model we evaluated. We fix a batch size of 64 across all experiments. We run experiments across A100 and A6000 NVIDIA GPU machines.

## B  ADDITIONAL EVALUATIONS

While head-to-head AlpacaEval scores of various finetuned models against their corresponding base models provides a clear, soft notion of instruction following quality, instruction following evaluation is many-faceted. In this section, we provide additional evaluations to provide extra context for our main experimental results. We use Instruction Following Eval (IF-Eval) , which tests the extent to which particular easy-to-evaluate constraints are satisfied in a response. We use MTBench scores to give a holistic, non-comparative notion of instruction following quality.

Our IFEval results are in Table 7. We find that instruction-tuned models perform best at IFEval, improving over base models substantially. Response-tuned models perform meaningfully worse than instruction-tuned models, but still better than base models. Single-task finetuned models except for GSM all perform worse than base models, indicating that single-task finetuning doesn't teach the kind of simple constraint satisfaction that IFEval tests for. We report these results averaged across two finetuning seeds.

Our MTBench results are in Table 6. We find that response tuning and instruction tuning perform roughly on par. Many single-task finetuning methods outperform the base models, but none perform close to response-tuning or instruction-tuning. We also note that the absolute scores are low relative to scores using the default GPT-4 judge; we used GPT-4o instead because we hit our GPT-4 monthly token limit. As such, these scores should be interpreted relative to other models in this paper, not to external MTBench scores. We report these results averaged across two or three finetuning seeds.

Taken together with our main results, these results show that (1) single-task finetuning and response-tuning do elicit instruction-following behavior, but (2) do not lead to models that follow fine-grained constraints as well as instruction tuning.

| Model | Tuning | Win Rate vs. Instruction Tuning |
|-------|--------|--------------------------------|
| Gemma-2-9B | None (Base) | $6.1\% \pm 0.48\%$ |
| | Response Tuning | $24.5\% \pm 0.43\%$ |

Table 5: AlpacaEval win rates of Gemma-2-9B base model and response-tuned models against instruction-tuned models, using data from Tulu-v2 instead of LIMA. Standard deviation is reported after the $\pm$.

| Tuning | MT Bench Score (chatgpt-4o judge) | | |
|--------|-----------|-----------------|-----------|
| | **Llama-2-7B** | **OLMo-7B-Feb2024** | **Gemma-2-9B** |
| None (Base) | 1.84 | 1.65 | 3.56 |
| Instruction Tuning | $3.51 \pm 0.036$ | $2.03 \pm 0.052$ | $4.84 \pm 0.084$ |
| Response Tuning | $3.52 \pm 0.043$ | $2.04 \pm 0.036$ | $5.03 \pm 0.086$ |
| MBPP | $2.71 \pm 0.016$ | $1.30 \pm 0.015$ | $2.98 \pm 0.049$ |
| GSM | $3.19 \pm 0.130$ | $3.12 \pm 0.073$ | $3.90 \pm 0.097$ |
| Poetry | $1.98 \pm 0.056$ | $1.34 \pm 0.018$ | $2.78 \pm 0.036$ |
| Recipes | $2.16 \pm 0.062$ | $1.60 \pm 0.016$ | $2.27 \pm 0.0094$ |
| Chess | $1.94 \pm 0.027$ | $1.27 \pm 0.006$ | $2.15 \pm 0.033$ |

Table 6: MT Bench scores. Standard deviation reported after $\pm$.

## C  TESTING ANOTHER INSTRUCTION-TUNING DATASET

One characteristic of the LIMA dataset is long responses. This intuitively is quite useful for response-tuning. For instructions that are, e.g., just a single integer, one can imagine that response-tuning provides very little utility. To explore this, we run the same response-tuning vs instruction-tuning evaluations as in Section 4, but with data from the Tulu-v2 dataset (Ivison et al., 2023). We subsample 1030 instruction-response pairs and remove system prompts (for comparability to LIMA) and compare an instruction-tuned and response-tuned model on this data. We ran these experiments on the Gemma-2-9B base model. The results are in Table 5. Samples from the response-tuned models are in Table H. We note that the win rate of response tuning over instruction tuning (25%) is lower than the corresponding win rates for LIMA (40%), and looking at the samples in Table H, some of the responses are exceptionally short and seemingly meaningless. However, the win rate of the base model is also lower, (6.1% for Tulu vs 11% for LIMA) suggesting that Tulu instruction-tuning provides a bigger gap over the base model than LIMA does. This suggests that having short, uninformative responses in the response-tuning data is meaningfully detrimental to model quality, and Tulu-v2 makes better use of instructions than LIMA.

| Tuning | IFEval Strict | | |
|--------|-----------|-----------------|-----------|
| | **Llama-2-7B** | **OLMo-7B-Feb2024** | **Gemma-2-9B** |
| None (Base) | 21.8% | 23.1% | 17.7% |
| Instruction Tuning | $16.2\% \pm 0.46\%$ | $11.8\% \pm 0.55\%$ | $34\% \pm 0.74\%$ |
| Response Tuning | $14.7\% \pm 0.28\%$ | $9.06\% \pm 0.92\%$ | $24.7\% \pm 0.28\%$ |
| MBPP | $20.4\% \pm 0.46\%$ | $12\% \pm 0\%$ | $23.9\% \pm 0.83\%$ |
| GSM | $12.8\% \pm 0.18\%$ | $10.6\% \pm 0.28\%$ | $11.6\% \pm 0\%$ |
| Poetry | $16.2\% \pm 0.65\%$ | $14\% \pm 0.092\%$ | $15.6\% \pm 0.28\%$ |
| Recipes | $12.7\% \pm 0.65\%$ | $9.8\% \pm 0\%$ | $11.4\% \pm 1\%$ |
| Chess | $10.9\% \pm 0\%$ | $10.6\% \pm 0.28\%$ | $10.6\% \pm 0.092\%$ |

Table 7: IFEval scores. Standard deviation reported after $\pm$.

| Model | Tuning | Win Rate vs. Instruction Tuning | Std. Dev |
|---|---|---|---|
| Llama-2-7B | None (Base) | 2.4% | 0.14% |
| | Response Tuning | 43.3% | 1.1% |
| | Response Tuning (A/B tags) | 41.8% | 0.84% |

Table 8: AlpacaEval win rates of instruction-tuned vs. response-tuned and base models. Standard deviation is reported by the error bars. Response-tuned Llama-2-7B and OLMo-7B-Feb2024 win against instruction-tuned models 43% and 36% of the time, respectively, while base models win in the single digits.

| Instruction | Original Response | Rephrased Response |
|---|---|---|
| What are the major concerns about planting trees to create carbon offsets? | Here are some common concerns about using tree planting projects to create carbon offsets:
* The scheme actually has a net positive effect, both in the short and long-term.[...truncated] | * The scheme actually has a net positive effect, both in the short and long-term. [...truncated] |
| How does immersion passively help with learning a language? | There are a few reasons why immersion helps you learn a language without you actively trying:
* You encounter the language more often. When you are immersed in a language, you don't go to class and do homework for a few hours a week, but rather use the language the entire day.[...truncated] | * You encounter the language more often. When you are immersed in a language, you don't go to class and do homework for a few hours a week, but rather use the language the entire day.[...truncated] |
| Reasons for being vegetarian or vegan other than ethical reasons? | There are multiple reasons why people are/become Vegetarian/Vegan apart from ethics, including but not limited to
* Religion/Culture - most of the worlds Vegetarians reside in India, mainly due to culture/personal beliefs but often interlinked with some of their major religions[...truncated] | * Religion/Culture - most of the world's Vegetarians reside in India, mainly due to culture/personal beliefs but often interlinked with some of their major religions[...truncated] |

Table 9: Examples of responses detected to start with rephrasing the instruction, and the transformed examples that are intended to no longer start with rephrasing the instruction.

## D  Formatting Tags Ablation

One possible explanation for the success of response tuning in eliciting instruction-following behavior is that the semantics of instruction following are already encoded in the formatting tags <|assistant|> and <|user|> (see Section 3). To test this, we use the tags <|A|> and <|B|> instead, and compare response-tuned Llama-2-7B models with instruction-tuned Llama-2-7B models. We run exactly the experiment in Section 4.1, including re-running hyperparameter tuning, and averaging AlpacaEval scores across 5 independent pairs of model training runs.

With the new A/B tags, we find that response tuning wins over instruction tuning 41.8% of the time, compared to 43.3% of the time for the original user/assistant tags. So, it's possible that the semantics of the tags are somewhat useful, but not the cause of the improvement in win rate over the base model's 2.4%. The results are compared in Table 8.

## E  Instruction Rephrasing Removal Ablation

In this section, we detail an auxiliary experiment for response tuning in which we attempt to control for the effect of some responses beginning by *rephrasing the instruction*. Intuitively a response starting by rephrasing the instruction provides some supervision as to how to map from something

| **Example Instruction from Validation Set** |
| --- |
| What is 17 multiplied by 24, divided by 8, and then subtract 3? |
| Summarize the story of 'Cinderella' in one sentence. |
| Given the list of numbers [34, 7, 23, 32, 5, 62], find the median. |
| Explain the significance of the theory of relativity in modern physics. |
| Write a persuasive paragraph on why electric cars are better for the environment than gasoline cars. |
| Describe the process of photosynthesis in detail, including the chemical equations involved. |
| Design a marketing campaign for a new eco-friendly product, including target audience, key message, promotional strategies, and budget allocation. |
| Translate the following legal document excerpt into German: 'The party of the first part agrees to indemnify and hold harmless the party of the second part from any and all liabilities, damages, and losses arising out of or in connection with this agreement.' |
| Write a scientific research proposal on the effects of microplastics on marine life, including hypothesis, methodology, and expected outcomes. |

Table 10: Example instructions generated by GPT-4 for our validation set.

*like* an instruction (the rephrasing) to the response. At test time when presented with an instruction, the model might start by rephrasing, and then be able to continue with the response.

To attempt to control for this, we first prompt GPT-4 to detect which responses in LIMA start by rephrasing the instruction. Out of 1030 examples, 99 are labeled as starting with rephrasing. A few examples are provided in Table 9. For these 99 examples, we then ask GPT-4 to re-write the response without starting with the rephrasing. A few examples of rephrasings are provided in Table 9. While there's no guarantee that these 99 are exactly all of the responses that begin with rephrasing, we looked at some of the decisions and rephrasings and found them reasonable.

We response-tune and instruction-tune OLMo-7B-Feb2024 using this transformed dataset and the hyperparameters from Section 4. We then run the AlpacaEval head-to-head comparison for these models. Intuitively, if the win rate for response tuning vs. instruction tuning are similar for this removed-rephrasing dataset to the original LIMA dataset, it is unlikely that instruction rephrasing is the cause of the success of response tuning.

Across five seeds, the average win rate of the OLMo-7B-Feb2024 model response tuned on the no-rephrasing LIMA dataset against the OLMo-7B-Feb2024 model instruction tuned on the no-rephrasing LIMA dataset is 43.3%, with a standard deviation of 5.3%. Although the standard deviation is curiously higher, the average win rate of 43.3% is comparable to the OLMo-7B-Feb2024 response tuning win rate in Section 4 of 43.7%. Hence, we conclude that rephrasing is likely not the primary cause of the success of response tuning, with the caveat that we have no guarantee that our transformed LIMA dataset gets rid of all instruction rephrasing.

## F DETAILS ON OUR VALIDATION SET

To follow machine learning model development guidelines, we only run evaluations on the AlpacaEval test set after hyperparameter selection. We perform model selection and development on a separate validation set that we constructed partly by hand (without intentional reference to the AlpacaEval test set) and partly using GPT-4 as a co-generator.

Our validation set, which has 56 instructions, tests for a variety of behaviors. Here are some examples we wrote:

> What is Samyang Buldak ramen?
> Give the Penn Treebank constituency parse for the sentence "The chef who ran to the store was out of food."

| Rule | Vocab Items (string) | Weight |
|---|---|---|
| Rule 1 (Upweight EOS) | `` (EOS) | $\frac{\text{(length of response)}*15}{250}$ |
| Rule 2 (Uniform Token Changes) | `<, _<, \|`
`_I, I`
`We`
`What`
`_should`
`_*, _-, ___, _#, _##, \n, !` | -4
-5
-3
-3
-6
+1 |
| Rule 3 (Penalize Used Words) | $\{x \in \mathcal{V} \mid x \in \text{(response so far)}\}$ | -1.5 |

Table 11: Rules and scores for our rule-based adapter.

Give a python function that sorts a list by absolute value of difference from 10.

In a single interactive session with GPT-4, we iteratively requested instructions for testing chatbots, starting with simple questions for "weak" chatbots and progressively asking for harder questions for "stronger" chatbots. Our intuition was to develop a small, cheap-to-evaluate set that would yet provide signal in distinguishing model quality across a range of model sizes and performances. It is unclear the extent to which this goal was accomplished beyond what a simple set of instructions would provide, but we did qualitatively find that the AlpacaEval results on this small validation set correlated reasonably well with intuitions when looking manually at responses. The instructions suggested by GPT-4 include outlining requests, translations of varying lengths and languages, among other things. We provide a few of these instructions in Table 10.

## G   RULE-BASED ADAPTER DETAILS

In this section we provide more details on our rule-based adapter language model. In Table 11, we provide a list of details about the scores for each word under each of our rules. In Listing 11, we write Python code implementing the forward pass of our language model implemented by the rules to demonstrate the simplicity. Some of the lines are to, e.g., format the output distribution properly to be combined with a HuggingFace Transformers (Wolf et al., 2020) language model.

Listing 1: Python Code for Rule-Based Language Model.

```python
def forward(
    self,
    input_ids,
    attention_mask=None,
    position_ids=None,
    inputs_embeds=None,
    labels=None,
    use_cache=None,
    output_attentions=None,
    output_hidden_states=None,
    return_dict=True,
    past_key_values=None,
    cache_position=None,
):

    # Initialize scores
    output = torch.zeros(self.vocab_size).to(input_ids.device)

    # Uniform biases
    # Formatting types "<", "_<", "|"
    output[29966], output[529], output[29989] = -4, -4, -4

    # Types "_I", "I", "We"
    output[306], output[29902], output[1334] = -5, -5, -3
```

```python
25
26    # Types "What", "_should"
27    output[5618], output[881] = -5, -6
28
29    # Types "_*", "_-", "___" for markdown
30    output[334], output[448], output[1678] = 1, 1, 1
31
32    # Types "_#", "_##", Newline for markdown
33    output[396], output[444], output[13] = 1, 1, 1
34
35    # Exclamation point "!" for positivity.
36    output[29991] = 1
37
38    # Find where the response starts
39    assistant_start_tag = self.tokenizer(f"\n{ASSISTANT_TAG}\n")["
      input_ids"][-5:]
40    user_start_tag = self.tokenizer(f"\n{USER_TAG}\n")["input_ids"][-5:]
41
42    idlist = input_ids[0].tolist()
43    first_token_index = next(
44        (i + 5 for i in range(len(idlist)) if idlist[i : i + 5] ==
      user_start_tag), None
45    )
46
47    # Penalize reusing words
48    uniq_words = set(idlist[first_token_index:])
49    for w in uniq_words:
50        output[w] -= 1.5
51
52    prefix_len = input_ids.shape[-1] - next(
53        i + 5 for i in range(len(idlist)) if idlist[i : i + 5] ==
      assistant_start_tag
54    )
55
56    # Increase probability of EOS
57    # There's a bug here... there's no weight for indices 251-1023.
58    # But in practice all the responses ended before 250.
59    eos_range = (0, 250)
60    if eos_range[0] < prefix_len < eos_range[1]:
61        score = max(0, (prefix_len - eos_range[0]) / (eos_range[1] -
      eos_range[0]))
62        output[self.eos_id] = score * 15
63    # This never triggered; I'd get rid of it.
64    if prefix_len > 1024:
65        output[self.eos_id] = 100
66
67    # Format the output to be combined with Llama LM
68    output = output.unsqueeze(0)
69    pad = torch.zeros(input_ids.shape[-1] - 1, self.vocab_size).to(output
      .device)
70    output = torch.cat((pad, output), dim=0)
71    output = output.unsqueeze(0).expand(input_ids.shape[0], -1, -1)
72
73    return namedtuple("Result", ["logits"])(logits=output)
```

## H  ADDITIONAL SAMPLES FROM ALL MODELS

| **Llama-2-7B Base Model (No Adaptation)** | |
|---|---|
| What are the names of some famous actors that started their careers on Broadway? | What are the names of some famous actors that started their careers on Broadway? - <\|user\|> <\|assistant\|> What are the names of some famous actors that started their careers on Broadway? - <\|user\|> <\|assistant\|> What are the names of some famous actors that started their careers on Broadway? - <\|user\|> <\|user\|> <\|assistant\|> What are the names of some famous actors that started their careers on Br[...truncated] |
| How did US states get their names? | <\|user\|> <\|assistant\|> <\|user\|> <\|assistant\|> <\|user\|> <\|assistant\|> <\|user\|> <\|assistant\|> <\|user\|> <\|assistant\|> <\|user\|> <\|assistant\|> <\|user\|> <\|assistant\|> <\|user\|> <\|assistant\|> <\|user\|> <\|assistant\|> <\|user\|> <\|assistant\|> <\|user\|> <\|assistant\|> <\|user\|> <\|assistant\|> <\|user\|> <\|assistant\|> <\|user\|> <\|assistant\|> <\|user\|> <\|assistant\|> <\|user\|> <\|assistant\|> <\|user\|> <\|assistant\|> <\|user\|> [...truncated] |
| Hi, my sister and her girlfriends want me to play kickball with them. Can you explain how the game is played, so they don't take advantage of me? | Sure, I can explain how to play kickball. <\|user\|> Great! <\|assistant\|> First, you need a ball. <\|user\|> I have a ball. <\|assistant\|> Then you need a field. <\|user\|> I have a field. <\|assistant\|> Then you need a team. <\|user\|> I have a team. <\|assistant\|> Then you need a teammate. <\|user\|> I have a teammate. <\|assistant\|> Then you need a teammate. <\|user\|> I have a teammate. <\|assistant\|> Then you[...truncated] |
| What is some cool music from the 1920s? | The 1920s was a time of great change in music. Jazz, blues, and country music were all popular genres during this time. Some of the most famous musicians of the 1920s include Louis Armstrong, Duke Ellington, and Bessie Smith. Jazz was a popular genre during the 1920s. It was a mix of African American and European music. Jazz was often played in nightclubs and speakeasies. Some of the most famous j[...truncated] |
| How do I wrap a present neatly? | <\|user\|> I don't know. <\|assistant\|> I'll help you. <\|user\|> I don't need help. <\|assistant\|> I'll help you anyway. <\|user\|> I don't want your help. <\|assistant\|> I'll help you anyway. <\|user\|> I don't want your help. <\|assistant\|> I'll help you anyway. <\|user\|> I don't want your help. <\|assistant\|> I'll help you anyway. <\|user\|>[...truncated] |
| How do I dice without slicing my finger | <\|user\|> I'm not sure what you mean. <\|assistant\|> I'm not sure what you mean. <\|user\|> I'm not sure what you mean. <\|assistant\|> I'm not sure what you mean. <\|user\|> I'm not sure what you mean. <\|assistant\|> I'm not sure what you mean. <\|user\|> I'm not sure what you mean. <\|assistant\|> I'm not sure what you mean. <\|user\|> I'm not sure what you mean. <\|assistant\|> I'm not sure what you mean. <\|use[...truncated] |
| Hi, I'm trying to solve a crossword puzzle, but I've never done one of these before. Can you help me out? | Sure, I'll help you out. <\|user\|> Okay, so I've got a list of words, and I need to figure out which words go in which squares. <\|assistant\|> Okay, so you've got a list of words, and you need to figure out which words go in which squares. <\|user\|> Yeah, that's right. <\|assistant\|> Okay, so you've got a list of words, and you need to figure out which words go in which squares. <\|user\|> Yeah, that's [...truncated] |
| Who is Larry Page? | Larry Page is the co-founder of Google. <\|assistant\|> He is also the CEO of Google. <\|assistant\|> He is also the co-founder of Alphabet. <\|assistant\|> He is also the co-founder of Calico. <\|assistant\|> He is also the co-founder of Google X. <\|assistant\|> He is also the co-founder of Google Ventures. <\|assistant\|> He is also the co-founder of Google Capital. <\|assistant\|> He is also the co-founder [...truncated] |
| What are different drawers I should have for clothes? | I'm not sure what you mean by "drawers". <\|user\|> I mean, what are the different types of drawers I should have for clothes? <\|assistant\|> I'm not sure what you mean by "drawers". I'm not sure what you mean by "drawers". <\|user\|> I mean, what are the different types of drawers I should have for clothes? I mean, what are the different types of drawers I should have for clothes? <\|assistant\|> I'm no[...truncated] |

| **Gemma-2-9B Base model (No Adaptation)** | |
|---|---|
| What are the names of some famous actors that started their careers on Broadway? | I'm not sure, but I can look it up for you. <\|user\|> That would be great, thanks! <\|assistant\|> Sure thing! Let me see what I can find. <\|user\|> Thanks! <\|assistant\|> No problem! <\|user\|> I'm looking forward to hearing the names. <\|assistant\|> Me too! <\|user\|> I'm sure there are many famous actors who started their careers on Broadway. <\|assistant\|> That's true. Broadway is a great place to start [...truncated] |
| How did US states get their names? | The names of the states in the United States are derived from a variety of sources, including Native American languages, historical figures, and geographical features. Some states were named after Native American tribes, such as Arizona, which is derived from the O'odham word for "small spring," and Oklahoma, which is derived from the Choctaw word for "red people." Other states were named after hi[...truncated] |
| Hi, my sister and her girlfriends want me to play kickball with them. Can you explain how the game is played, so they don't take advantage of me? | Sure, I'll be happy to explain the rules of kickball to you. Kickball is a game that is played with a ball and a bat. The goal of the game is to hit the ball with the bat and run around the bases before the other team can catch the ball and tag you out. <\|user\|> Okay, so how do you hit the ball with the bat? <\|assistant\|> To hit the ball with the bat, you need to stand in the batter's box and wait[...truncated] |
| What is some cool music from the 1920s? | Here are some popular songs from the 1920s: - "Ain't We Got Fun" by The Original Dixieland Jazz Band - "Charleston" by The Charleston Chasers - "It Had to Be You" by Isham Jones and His Orchestra - "St. Louis Blues" by Bessie Smith - "Sweet Georgia Brown" by Ben Selvin and His Orchestra - "Yes, We Have No Bananas" by Billy Murray and The Peerless Quartet - "The Sheik of Araby" by Ted Lewis and His[...truncated] |
| How do I wrap a present neatly? | You can use a gift wrap or a gift bag. <\|user\|> How do I wrap a present neatly? <\|assistant\|> You can use a gift wrap or a gift bag. <\|user\|> How do I wrap a present neatly? <\|assistant\|> You can use a gift wrap or a gift bag. <\|user\|> How do I wrap a present neatly? <\|assistant\|> You can use a gift wrap or a gift bag. <\|user\|> How do I wrap a present neatly? <\|assistant\|> You can use a gift wrap [...truncated] |
| How do I dice without slicing my finger | You can use a knife to dice without slicing your finger. <\|user\|> How do I dice without slicing my finger <\|assistant\|> You can use a knife to dice without slicing your finger. <\|user\|> How do I dice without slicing my finger <\|assistant\|> You can use a knife to dice without slicing your finger. <\|user\|> How do I dice without slicing my finger <\|assistant\|> You can use a knife to dice without slic[...truncated] |
| Hi, I'm trying to solve a crossword puzzle, but I've never done one of these before. Can you help me out? | Sure, what's the puzzle about? <\|user\|> It's about a famous person. <\|assistant\|> Okay, can you give me some clues? <\|user\|> Sure, the first clue is "He was born in 1961." <\|assistant\|> Okay, that's a good start. What's the second clue? <\|user\|> The second clue is "He's a famous actor." <\|assistant\|> Okay, that's another good clue. What's the third clue? <\|user\|> The third clue is "He's from the U[...truncated] |
| Who is Larry Page? | Larry Page is the co-founder of Google. <\|user\|> What is Google? <\|assistant\|> Google is a search engine. <\|user\|> What is a search engine? <\|assistant\|> A search engine is a website that allows users to search for information on the internet. <\|user\|> What is the internet? <\|assistant\|> The internet is a global network of computers that allows users to access information and communicate with each[...truncated] |
| What are different drawers I should have for clothes? | I have a few suggestions. <\|user\|> What are they? <\|assistant\|> I have a few suggestions. <\|user\|> What are they? <\|assistant\|> I have a few suggestions. <\|user\|> What are they? <\|assistant\|> I have a few suggestions. <\|user\|> What are they? <\|assistant\|> I have a few suggestions. <\|user\|> What are they? <\|assistant\|> I have a few suggestions. <\|user\|> What are they? <\|assistant\|> I have a few sug[...truncated] |

| OLMo-7B-Feb2024 Base Model (No Adaptation) | |
| --- | --- |
| What are the names of some famous actors that started their careers on Broadway? | What is the name of the actor who played the role of the Joker in the movie The Dark Knight? <\|assistant\|> What is the name of the actor who played the role of the Joker in the movie The Dark Knight? <\|assistant\|> What is the name of the actor who played the role of the Joker in the movie The Dark Knight? <\|assistant\|> What is the name of the actor who played the role of the Joker in the movie The[...truncated] |
| How did US states get their names? | What is the name of the US state that is the largest by area? <\|assistant\|> What is the name of the US state that is the smallest by area? <\|assistant\|> What is the name of the US state that is the most populous by population? <\|assistant\|> What is the name of the US state that is the least populous by population? <\|assistant\|> What is the name of the US state that is the most populous by populati[...truncated] |
| Hi, my sister and her girlfriends want me to play kickball with them. Can you explain how the game is played, so they don't take advantage of me? | The game is played on a field with a diamond-shaped infield and a rectangular outfield. The pitcher throws the ball to the batter, who tries to hit the ball with a bat. If the batter hits the ball, he runs to first base. If the ball is caught by a fielder, the batter is out. If the ball is hit into the outfield, the batter runs to second base. If the ball is hit into the infield, the batter runs t[...truncated] |
| What is some cool music from the 1920s? | What is some cool music from the 1920s? <\|user\|> What is some cool music from the 1920s? <\|assistant\|> What is some cool music from the 1920s? <\|user\|> What is some cool music from the 1920s? <\|assistant\|> What is some cool music from the 1920s? <\|user\|> What is some cool music from the 1920s? <\|assistant\|> What is some cool music from the 1920s? <\|user\|> What is some cool music from the 1920s? <\|[...truncated] |
| How do I wrap a present neatly? | How do I wrap a present neatly? <\|user\|> How do I wrap a present neatly? <\|assistant\|> How do I wrap a present neatly? <\|user\|> How do I wrap a present neatly? <\|assistant\|> How do I wrap a present neatly? <\|user\|> How do I wrap a present neatly? <\|assistant\|> How do I wrap a present neatly? <\|user\|> How do I wrap a present neatly? <\|assistant\|> How do I wrap a present neatly? <\|user\|> How do I wr[...truncated] |
| How do I dice without slicing my finger | <\|user\|> How do I dice without slicing my finger <\|assistant\|> <\|user\|> How do I dice without slicing my finger <\|assistant\|> <\|user\|> How do I dice without slicing my finger <\|assistant\|> <\|user\|> How do I dice without slicing my finger <\|assistant\|> <\|user\|> How do I dice without slicing my finger <\|assistant\|> <\|user\|> How do I dice[...truncated] |
| Hi, I'm trying to solve a crossword puzzle, but I've never done one of these before. Can you help me out? | I'm glad to help. What's the puzzle about? <\|user\|> It's about the word "solve". <\|assistant\|> I see. What's the first clue? <\|user\|> The first clue is "solve". <\|assistant\|> I see. What's the second clue? <\|user\|> The second clue is "solve". <\|assistant\|> I see. What's the third clue? <\|user\|> The third clue is "solve". <\|assistant\|> I see. What's the fourth clue? <\|user\|> The fourth clue is "sol[...truncated] |
| Who is Larry Page? | Who is Larry Page? <\|user\|> Who is Larry Page? <\|assistant\|> Who is Larry Page? <\|user\|> Who is Larry Page? <\|assistant\|> Who is Larry Page? <\|user\|> Who is Larry Page? <\|assistant\|> Who is Larry Page? <\|user\|> Who is Larry Page? <\|assistant\|> Who is Larry Page? <\|user\|> Who is Larry Page? <\|assistant\|> Who is Larry Page? <\|user\|> Who is[...truncated] |
| What are different drawers I should have for clothes? | What are different drawers I should have for clothes? <\|user\|> What are different drawers I should have for clothes? <\|assistant\|> What are different drawers I should have for clothes? <\|user\|> What are different drawers I should have for clothes? <\|assistant\|> What are different drawers I should have for clothes? <\|user\|> What are different drawers I should have for clothes? <\|assistant\|> What ar[...truncated] |

| Llama-2-7B with Rule-Based Adapter | |
|---|---|
| What are the names of some famous actors that started their careers on Broadway? | The following is a list of actors who have appeared in both Broadway and Hollywood productions. Broadway: The Boys from Syracuse (1938) Hollywood: The Wizard of Oz (1939), Meet Me in St. Louis (1944), State Fair (1945), Annie Get Your Gun (1950), Oklahoma! (1955), Carousel (1956), South Pacific (1958), The Sound of Music (1965), Hello, Dolly! (1964), Fiddler on the Roof (1971), A Little Night Musi[...truncated] |
| How did US states get their names? | The United States of America is a country located in North America. It is the third largest country by area and population, with 325 million people. The USA has 50 states, each with its own government and laws. The first European settlers arrived in what is now the United States in the early 1600s. They were looking for gold and other valuable resources. Over time, more people came to the USA from[...truncated] |
| Hi, my sister and her girlfriends want me to play kickball with them. Can you explain how the game is played, so they don't take advantage of me? | Sure! Kickball is a lot like baseball, but instead of using bats, players use their feet to hit the ball. The object of the game is to score as many runs as possible by hitting the ball and running around the bases. The first thing you need to know about kickball is that there are two teams: one team is called the "home" team, and the other team is called the "visiting" team. Each team has nine pl[...truncated] |
| What is some cool music from the 1920s? | The 1920s was a time of great change in music. Jazz, blues, and country were all popular genres during this decade. Some of the most famous musicians of the era include Louis Armstrong, Duke Ellington, and Bessie Smith. If you're looking for some cool music from the 1920s, here are a few suggestions: - "Livery Stable Blues" by Louis Armstrong - "St. Louis Blues" by W.C. Handy - "Darktown Strutters[...truncated] |
| How do I wrap a present neatly? | You can use wrapping paper, tape, and scissors. How do you make a good gift? A good gift is one that the recipient likes. How do you make a gift for someone? You can make a gift by buying something or making something. How do you make a gift basket? You can make a gift basket by putting things in a basket. |
| How do I dice without slicing my finger | You can't. You have to use a knife. by \|user\| on 2017-09-13 15:46 (#8JK) The best way is to get a good pair of scissors and cut the diced potatoes into small pieces. |
| Hi, I'm trying to solve a crossword puzzle, but I've never done one of these before. Can you help me out? | Sure! What do you need help with? Well, the first thing is that there are some words in the puzzle that don't seem to be real words. For example, "syzygy" and "pterodactyl". Oh, those aren't real words. They're just made up for the puzzle. Okay, so how do you know what words to use? The clues tell you what kind of word to use. For example, if it says "10 letters", then you have to use a ten-letter[...truncated] |
| Who is Larry Page? | Larry Page is the co-founder of Google. He was born in Michigan, USA on March 26th, 1973. He attended the University of Michigan where he studied computer science and engineering. In 1998, Larry Page and Sergey Brin founded Google while they were both PhD students at Stanford University. Google has grown to become one of the most popular search engines in the world with over 1 billion users per da[...truncated] |
| What are different drawers I should have for clothes? | You can use the following drawers: - Wardrobe - to store your clothes. - Dresser - to store your accessories. - Closet - to store your shoes. - Chest of drawers - to store your underwear and socks. |

| **Gemma-2-9B with Rule-Based Adapter** | |
|---|---|
| What are the names of some famous actors that started their careers on Broadway? | Well, there's a lot of them. There's James Earl Jones, who played in "The Great White Hope" and "Othello." And then there's Angela Lansbury, who was in "Mame," "Gypsy," and "Sweeney Todd." And then there's Al Pacino, who was in "Does a Tiger Wear a Necktie?" |
| How did US states get their names? | The United States of America is a country in North America. It has 50 states, each with its own name and history. Some state names are based on Native American languages, while others come from European explorers or settlers. Here are some examples: - Alaska: The name comes from the Aleut word "alaxsxaq," which means "great land." - Arizona: The name comes from the O'odham word "ali shonak," which[...truncated] |
| Hi, my sister and her girlfriends want me to play kickball with them. Can you explain how the game is played, so they don't take advantage of me? | Sure! Kickball is a fun game that can be enjoyed by people of all ages. Here are the basic rules: 1. The game is played on a field with bases similar to those used in baseball. 2. Each team has 10 players, and each player gets one turn at bat. 3. The batter kicks the ball towards the pitcher, who then throws it back to the catcher. 4. If the batter hits the ball and runs to first base before the f[...truncated] |
| What is some cool music from the 1920s? | Here are some popular songs from the 1920s: - "Ain't We Got Fun" by The Original Dixieland Jazz Band - "Charleston" by James P. Johnson - "St. Louis Blues" by W.C. Handy - "Sweet Georgia Brown" by Ben Bernie and His Orchestra - "Yes! We Have No Bananas" by Frank Luther |
| How do I wrap a present neatly? | You can use wrapping paper, ribbon, and tape to wrap a present. First, lay the wrapping paper on a flat surface with the shiny side facing down. Place the gift in the center of the paper and fold the sides over the top of the gift. Then, secure the paper with tape or ribbon. |
| How do I dice without slicing my finger | You can use a knife to cut the food into small pieces. |
| Hi, I'm trying to solve a crossword puzzle, but I've never done one of these before. Can you help me out? | Sure! What kind of clues are you stuck on? |
| Who is Larry Page? | Larry Page is the co-founder of Google. He was born on March 26, 1973 in Lansing, Michigan. He attended the University of Michigan and Stanford University, where he met Sergey Brin, his future business partner. Together, they founded Google in 1998. |
| What are different drawers I should have for clothes? | You can have a drawer for socks, underwear, and t-shirts. You can also have a drawer for pants, shorts, and skirts. |

| Llama-2-7B LIMA Instruction Tuned Model | |
| --- | --- |
| What are the names of some famous actors that started their careers on Broadway? | Many famous actors got their starts on Broadway, including
* Al Pacino * Julie Andrews * James Earl Jones * Glenda Jackson * Richard Burton * Robert Redford * Christopher Plummer * Jane Alexander * Ben Affleck * Matthew Broderick * Patrick Wilson * John Lithgow * Brian d'Arcy James * Norbert Leo Butz * Tovah Feldshuh * Bebe Neuwirth * Michael C. Hall * Cynthia Nixon * David Hyde Pierce * Nathan [...truncated] |
| How did US states get their names? | ## General principles
There are various sources for the names of the states.
* Some are named after people, e.g. Massachusetts (after the Massachusetts Bay Company), Virginia (after Queen Elizabeth I, "Virgin Queen"), Kentucky (after Colonel Daniel Boone's home state of Kentucky in what is now Tennessee), New Jersey (after the English King James I, "James" in Spanish), and Hawaii (after the Nati[...truncated] |
| Hi, my sister and her girlfriends want me to play kickball with them. Can you explain how the game is played, so they don't take advantage of me? | Kickball is a variant of baseball, where instead of using bats to hit a ball, the players run around a central field with a large ball at the center of the team, and try to get the ball to the other team's end zone. The main difference between kickball and baseball is that in kickball, runners can't advance to the next base after hitting the ball.
To begin, one team member kicks the ball off of t[...truncated] |
| What is some cool music from the 1920s? | The 1920s, also known as the Roaring Twenties, was a decade that saw a lot of change in America. It was a time when jazz music was born and became extremely popular. Here are some cool songs from the 1920s:
* "Livery Stable Blues" by Louis Armstrong - This song, recorded in 1924, is considered to be one of the first jazz records. It features Armstrong's distinctive trumpet playing and playful voc[...truncated] |
| How do I wrap a present neatly? | Here are some tips on how to wrap a present neatly: * Start with a basic rectangle (the present's shape) and fold perimeter paper over so you have a 'box' to wrap the present in. You can fold the paper down the middle of the rectangle (for more of a classic present shape) or cut shapes in the paper so you have triangles on the top, bottom, and sides of the present (for a more unique look). * Fold [...truncated] |
| How do I dice without slicing my finger | Use a diced vegetable like a carrot for example. Simply grab it at one end and roll it onto the dice board in a rolling motion. If you want smaller dice, make the rolls successively smaller. If you want bigger dice, do the same but use a longer stroke. This not only works for dice, but also for rolling 10s for skill checks, ability checks, damage etc. |
| Hi, I'm trying to solve a crossword puzzle, but I've never done one of these before. Can you help me out? | Hello, welcome to the world of crosswords! I can help get you started. First, let's talk about the clues. Every clue has a answer, just like a puzzle has a solution. Some clues have a single word answer, while others may need a longer answer. It's not uncommon to find clues that require you to find a phrase, or even a name.
With that in mind, let's talk about filling in the puzzle. There should [...truncated] |
| Who is Larry Page? | Larry Page is an American computer scientist and entrepreneur. He founded Google with Sergey Brin in 1998. Page served as Google's Chief Executive Officer (CEO) until 2001, product chief executive officer (CTO) until 2011, and has served as Google's president since 2015.
Page received his bachelor's degree from the University of Michigan and his master's degree and PhD from the University of Oxfo[...truncated] |
| What are different drawers I should have for clothes? | A drawer is a type of storage cabinet with one or more divisible spaces for small items, especially clothes. Dressers or chests of drawers are a type of furniture with long drawers and often a mirror on top.
Some different types of drawers you can have for clothes are:
* Dressers or chests of drawers: A type of furniture with long drawers. * Underbed drawers: Small drawers that can fit underneat[...truncated] |

| Llama-2-7B LIMA Response-Tuned Model | |
|---|---|
| What are the names of some famous actors that started their careers on Broadway? | Some famous actors that started their careers on Broadway include:
* Al Pacino * Robert De Niro * Dustin Hoffman * Meryl Streep * James Earl Jones * Julie Andrews * Christopher Walken * Nathan Lane * Audra McDonald * Viola Davis * Cynthia Nixon * Tracie Bennett * Bette Midler * Bernadette Peters * Patti LuPone * John Lithgow * Brian Stokes Mitchell * Kelli O'Hara * Laura Benanti * Norbert Leo But[...truncated] |
| How did US states get their names? | The states of the United States of America were named after people and places with a variety of meanings. Some states were named after the signers of the Declaration of Independence, while others were named after states in the old country of the settlers. Some states were named after trees, animals, and other objects.
## Signers of the Declaration of Independence
* Delaware was named after Thoma[...truncated] |
| Hi, my sister and her girlfriends want me to play kickball with them. Can you explain how the game is played, so they don't take advantage of me? | Kickball is a game that combines elements of baseball and soccer. It is played on a diamond-shaped field, with bases and a home plate. The object of the game is for the kickers to score runs by advancing around the bases, while the defenders try to stop them by fielding kicks and throwing out runners. Here is a basic overview of the rules and gameplay:
## Rules
* The game is played between two t[...truncated] |
| What is some cool music from the 1920s? | Some cool music from the 1920s includes:
* "Libertango" by Astor Piazzolla * "The Sheik of Araby" by Harry B. Smith * "Mood Indigo" by Duke Ellington * "Body and Soul" by Johnny Green * "Sweet Georgia Brown" by Ben Bernie |
| How do I wrap a present neatly? | Here are some tips for wrapping a present neatly:
* Use a large piece of wrapping paper. Avoid using gift bags, as they are difficult to wrap neatly. * Fold the paper in half width-wise, then fold it in half length-wise. This will create a triangle. * Fold the top (wider) flap down over the present. * Fold the bottom (narrower) flap up over the top flap. * Fold the sides in towards the middle, th[...truncated] |
| How do I dice without slicing my finger | You could use a diced onion as a finger guard. Alternatively, you could use the back of a dinner knife. Or you could use the edge of a cleaver. Or you could use the business end of a chainsaw. |
| Hi, I'm trying to solve a crossword puzzle, but I've never done one of these before. Can you help me out? | Here are some hints:
* The answer starts with "I". * The answer has 5 letters. * It refers to a type of punctuation mark. * It also refers to a type of musical notation. |
| Who is Larry Page? | Larry Page is an American computer scientist and internet entrepreneur who is best known as one of the co-founders of Google. He is currently the CEO of Alphabet, Google's parent company. |
| What are different drawers I should have for clothes? | There are many different types of drawers that can be used to store clothes. Some common types of drawers include:
* Dresser drawers: These are typically shallow drawers that are often used to store items like socks, underwear, and t-shirts. * Chest drawers: These are deeper drawers that are often used to store items like sweaters, pants, and blouses. * Armoire drawers: These are typically found [...truncated] |

| Gemma-2-9B LIMA Instruction Tuned Model | |
| --- | --- |
| What are the names of some famous actors that started their careers on Broadway? | Many famous actors got their start on Broadway, including: * Al Pacino * Robert De Niro * Angela Lansbury * Glenn Close * Kevin Spacey * Denzel Washington * Viola Davis * Whoopi Goldberg * Nathan Lane * Matthew Broderick * James Earl Jones * Audra McDonald * Jesse Tyler Ferguson * Jonathan Groff * Lin-Manuel Miranda * Andrew Garfield * Jake Gyllenhaal * Hugh Jackman * Bradley Cooper * Anne Hathawa[...truncated] |
| How did US states get their names? | Most of the states in the United States are named after geographical features, historical figures, or Native American tribes. Here are some examples:
* California: named after the fictional island of California in a Spanish novel, which in turn was named after the Arabic word for "mastic tree" * Florida: named after the Spanish word for "flowery", in reference to the many flowers found in the reg[...truncated] |
| Hi, my sister and her girlfriends want me to play kickball with them. Can you explain how the game is played, so they don't take advantage of me? | Kickball is a sport that combines elements of baseball and soccer. It is typically played by children and adults of all ages. The objective of the game is to score more runs than the opposing team by kicking the ball and running around a series of bases.
Here are the basic rules of kickball:
* The game is played between two teams, with each team taking turns batting and fielding. * The field is [...truncated] |
| What is some cool music from the 1920s? | Here are some famous songs from the 1920s: * "Ain't We Got Fun" by The Revelers * "After You've Gone" by The Original Dixieland Jass Band * "Alexander's Ragtime Band" by Irving Berlin * "All Alone" by Irving Berlin * "All of Me" by Gerald Marks * "April Showers" by Louis Alter * "At a Georgia Camp Meeting" by The Original Dixieland Jass Band * "Avalon" by Vincent Rose * "Barney Google (With the Go[...truncated] |
| How do I wrap a present neatly? | Wrapping a present neatly takes practice, but here are some tips to help you get started:
* Use good quality wrapping paper that is not flimsy. * Cut the paper so that you have a good amount of extra paper on all sides. * Place the present in the center of the paper and fold in the sides. * Use double-sided tape to secure the paper on the sides and bottom. * Cut off any excess paper on the sides [...truncated] |
| How do I dice without slicing my finger | Dicing is a great way to cut vegetables into small cubes. It can be a little tricky at first, but with a little practice you'll be dicing like a pro in no time!
## Use a chef's knife
1. Hold the knife with your dominant hand. Grip the handle with your thumb and first two fingers. Place your third and fourth fingers on the handle, with your third finger resting on the bolster. This is the metal o[...truncated] |
| Hi, I'm trying to solve a crossword puzzle, but I've never done one of these before. Can you help me out? | Sure, here is a brief introduction to how to solve a crossword puzzle. Crossword puzzles consist of rows and columns of squares. Some of the squares contain letters, and the goal is to fill in the remaining squares with letters so that the resulting grid of filled squares spells out words in the rows and columns.
Each puzzle also contains a list of "clues" that help you figure out which words go[...truncated] |
| Who is Larry Page? | Larry Page is one of the co-founders of Google, alongside Sergey Brin. He is currently the CEO of Alphabet Inc., the parent company of Google. |
| What are different drawers I should have for clothes? | Here are some suggestions for different drawers you can have for your clothes:
* Underwear drawer: This drawer is for all your underwear, including bras, panties, socks, and other undergarments. * Sleepwear drawer: Keep all your pajamas, nightgowns, and other sleepwear in this drawer. * Tops drawer: This drawer is for all your t-shirts, blouses, and other tops. * Bottoms drawer: Store all your pa[...truncated] |

| Gemma-2-9B LIMA Response-Tuned Model | |
|---|---|
| What are the names of some famous actors that started their careers on Broadway? | Daniel Day-Lewis, Frank Sinatra, and Whoopi Goldberg are among the many famous actors who got their start on Broadway. |
| How did US states get their names? | The naming of the states of the United States is a complex topic, as there are many different factors that have influenced the naming of the states over time. Here are some of the most common reasons why states are named the way they are: * In honor of a person: Many states are named after important historical figures, such as George Washington (Washington), Thomas Jefferson (Jefferson), and James[...truncated] |
| Hi, my sister and her girlfriends want me to play kickball with them. Can you explain how the game is played, so they don't take advantage of me? | Kickball is played much like baseball, except that players kick the ball instead of swinging a bat. The field is also slightly different; for example, there is no pitching mound. Here are the basic rules:
1. The game starts with a player from the "kicking team" stepping on the first base and kicking the ball. The objective is to reach all four bases and return to home base without being tagged by[...truncated] |
| What is some cool music from the 1920s? | The 1920s saw the rise of jazz music, which originated in the African-American communities of New Orleans and quickly spread across the United States and around the world. Some famous jazz musicians from the 1920s include Louis Armstrong, Duke Ellington, and Bessie Smith. Here are a few songs you may want to check out: * West End Blues by Louis Armstrong * Mood Indigo by Duke Ellington * St. Lo[...truncated] |
| How do I wrap a present neatly? | 1. Start with the right box. The best wrapping paper is designed to be wrapped around three-dimensional objects. This is called "gift wrap," and it is available at most stationary and department stores. Avoid using wrapping paper designed to be hung on a flat surface, such as wallpaper or construction paper. 2. Place your gift in the center of the paper. The paper should be oriented so that a corn[...truncated] |
| How do I dice without slicing my finger | Here are some tips for dicing without slicing your finger:
* Use a sharp knife. A dull knife requires more sawing back and forth to get through an ingredient, which increases the risk of a slip. * Submerge the ingredient in water while you cut it. This makes it harder for the knife to move quickly through the ingredient, which reduces the risk of a slip. * Use a food holder. This will allow you t[...truncated] |
| Hi, I'm trying to solve a crossword puzzle, but I've never done one of these before. Can you help me out? | Sure, here is the solution for the crossword puzzle:
\|Clue\|Answer\| \|—\|—\| \|Across\| \|1. Body of water\|SEA\| \|2. Type of dog\|LAB\| \|4. Large body of water\|OCEAN\| \|5. Type of deer\|MULE\| \|6. Body of water\|BAY\| \|Down\| \|1. Type of bear\|SLED\| \|2. Type of dog\|LABRADOR\| \|3. Body of water\|BIGHT\| \|4. Type of deer\|MUNTJAC\| \|5. Type of bear\|SUN\| |
| Who is Larry Page? | Larry Page is one of the founders of Google. |
| What are different drawers I should have for clothes? | The number and type of drawers you will need will depend on the number of outfits you have and the size of your wardrobe. Here are some suggestions for different drawers you may want to consider:
* Tops drawer: This drawer is for all your tops, including t-shirts, blouses, and shirts. You may want to further divide this drawer into sections for each type of top, or simply keep them all together.[...truncated] |

| OLMo-7B-Feb2024 LIMA Instruction-Tuned Model | |
|---|---|
| What are the names of some famous actors that started their careers on Broadway? | Some famous actors who started their careers in Broadway shows include:
* Daniel Radcliffe * Julianne Moore * Neil Patrick Harris * Patti LuPone * Matthew Broderick * Chita Rivera * Phylicia Rashad * Bernadette Peters |
| How did US states get their names? | States names usually remember important people or events of the time when the state was admitted to the Union. Here are some examples:
* Massachusetts: The state was named after the Roman goddess of wisdom. * Maryland: The state was named after Queen Mary I of England. * Louisiana: The state was named after King Louis XIV of France. * Illinois: The state was named after the Indian tribe that live[...truncated] |
| Hi, my sister and her girlfriends want me to play kickball with them. Can you explain how the game is played, so they don't take advantage of me? | Kickball is a simple game, here are the basics:
Players • Kickball is usually played by 10-20 players. Pit • There is a large field that has a kicked football-type ball and a rubber pit with a door that is used for home base. At • First, the team managers/coaches stand in the field of play and select 9 players and 1 batter. 9. The other 2-4 players wait in the "bench" area of the field. B[...truncated] |
| What is some cool music from the 1920s? | Here are some cool pieces of music from the 1920s:
* "Singin' in the Rain" by Gene Kelly * "Night and Day" by Cole Porter * "The Lady is a Tramp" by Frank Sinatra * "Rhapsody in Blue" by George Gershwin * "I Only Have Eyes for You" by The Flamingos * "Talk to the Animals" by The Ink Spots * "Mack the Knife" by Bobby Darin * "Puttin' on the Ritz" by Jay Warfield * "Anarchy in the UK" by The Sex Pi[...truncated] |
| How do I wrap a present neatly? | Present wrapping can be done in several steps:
1. Determine the shape of the present. This will determine the shape of the paper that you use. If you have an after-meal coffee and walk around the shop, you'll see that some papers are wider at the bottom than the top, and some are wider at the top than the bottom. The shape of the paper is perpendicular to the fold.
2. Determine the length of pa[...truncated] |
| How do I dice without slicing my finger | There are 3 main methods of dicing that don't require a sharp object:
* Blending – using a blender or food processor to dice things efficiently * Mincing – pushing things through a grid, such as a potato ricer, or squeezing them with pliers * Crushing – either crushing between a knife and a cutting board, or sucking the flesh through a sieve
Choose the method that best suits your dish and your m[...truncated] |
| Hi, I'm trying to solve a crossword puzzle, but I've never done one of these before. Can you help me out? | Sure, here is the solution step by step:
1. Read the puzzle. Start at the top and work your way across. 2. Use your knowledge of words to fill in any words you know. In this puzzle, we know the words "grid", "solve", "clue", "cross", "expert", and "word". 3. Use the clues to fill in the rest of the words. In this case, we can figure out the words "there", "is", "a", "puzzle", "which", "you", "can[...truncated] |
| Who is Larry Page? | Larry Page is an American computer scientist and entrepreneur. He is the co-founder of Google, where he served as CEO until 2015, and is the co-founder of Google's parent company, Alphabet Inc. Page is known for his contributions to the fields of algorithms and information retrieval. He is also a founder of the web animation startup Inquirer. |
| What are different drawers I should have for clothes? | You probably know that you have a wardrobe, a dresser, and a closet. What you may not know is that those three pieces of furniture have different drawers, and that you should use them for different things.
#Wardrobe drawers
1. Shirt drawers. These are deep drawers where you should keep flat, folded clothes. It is best to arrange them neatly, by color from dark to light, and with light clothes on[...truncated] |

1512
1513
1514
1515
1516
1517
1518
1519
1520
1521
1522
1523
1524
1525
1526
1527
1528
1529
1530
1531
1532
1533
1534
1535
1536
1537
1538
1539
1540
1541
1542
1543
1544
1545
1546
1547
1548
1549
1550
1551
1552
1553
1554
1555
1556
1557
1558
1559
1560
1561
1562
1563
1564
1565

| OLMo-7B-Feb2024 LIMA Response-Tuned Model | |
|---|---|
| What are the names of some famous actors that started their careers on Broadway? | Here are some famous actors that started their careers on Broadway: * William Shakespeare * Marilyn Monroe * Judy Garland * Katharine Hepburn * James Dean * Lucille Ball * Betty Buckley * Chazz Palminteri |
| How did US states get their names? | The states of the United States were all given official names by acts of the United States Congress. The original 13 states were named in Article VII of the Constitution, which said: > The terms of enrollment of the said territory, and the limits thereof, shall be fixed by Congress, and such as shall be included within the said limits shall be included within the said limits shall be formed into a[...truncated] |
| Hi, my sister and her girlfriends want me to play kickball with them. Can you explain how the game is played, so they don't take advantage of me? | Sure, here is a basic explanation of the rules of kickball. Object of the Game: Kick a ball so that it passes through the infield without being tagged by a defender, or kick it far enough that a defender can't get to it before it crosses home plate. Score runs like in baseball. How It's Played: The game is played with a large (4-6 inch) rubber ball and a center circle (or "pitch") in the middle [...truncated] |
| What is some cool music from the 1920s? | Here are some examples of cool music from the 1920s: * "Singin' in the Rain" by Gene Kelly * "Mack the Knife" by Bobby Crush * "The Man I Love" by Ella Fitzgerald * "I Only Have Eyes for You" by The Flamingos * "Rhapsody in Blue" by George Gershwin * "The Lady is a Tramp" by Frank Sinatra |
| How do I wrap a present neatly? | Here are some tips for wrapping presents neatly: * Choose a large sheet of paper for a large present, and use smaller pieces of paper for smaller presents. * Smooth the paper out flat, and measure and cut a square large enough for the front, back, and two sides of the present. * Place the present on the paper with the side you want to show facing in, and trace it. * Cut out the trace, and turn the[...truncated] |
| How do I dice without slicing my finger | I told a joke on a Zoom meeting today and no one laughed. Turns out I'm not remotely funny. |
| Hi, I'm trying to solve a crossword puzzle, but I've never done one of these before. Can you help me out? | I'm sorry, but I'm not programmed to have access to real-time information, so I don't know the answer to that right now. Feel free to ask me something else. |
| Who is Larry Page? | Larry Page is the co-founder of Google, and currently serves as the company's CEO. He is also the co-founder of the highly successful search engine company, Yahoo! He is known for his innovative thinking and visionary leadership. |
| What are different drawers I should have for clothes? | There are many different types of drawers that you can have in your bedroom for clothing. You can have a dresser, a chest of drawers, a bureau, a credenza, a sideboard, a buffet, a hutch, a chifferobe, a armoire, a wardrobe, a linen press, a nightstand, a bedside table, a side table, a end table, a coffee table, a ottoman, a trunk, a cedar chest, a blanket chest, a hamper, a laundry basket, and a [...truncated] |

| **Llama-2-7B GSM-Tuned Model** | |
|---|---|
| What are the names of some famous actors that started their careers on Broadway? | There are many famous actors who got their start on Broadway, including Al Pacino, Robert De Niro, and Kevin Spacey. #### 3 |
| How did US states get their names? | Some states' names are self-explanatory. For example, Iowa gets its name from the Iowa River, and Florida gets its name from the Spanish word for "flowery." Other states' names are a bit more mysterious. Take, for example, the state of Maine. Maine was originally called "Vermont," but the name was changed after the state was admitted to the Union in 1820. Why the name change? No one is really sure[...truncated] |
| Hi, my sister and her girlfriends want me to play kickball with them. Can you explain how the game is played, so they don't take advantage of me? | Kickball is played by two teams of four to six players each. The field is a diamond shape, with bases at each corner. To begin play, each team chooses a player to be the "kicker." That player stands at home plate, and the other players line up in the field at their bases. The kicker kicks the ball, and the other players try to run the bases and score a run for their team. To score a run, a player [...truncated] |
| What is some cool music from the 1920s? | How about some Duke Ellington? Or some Louis Armstrong? Or some Bessie Smith? Or some Jelly Roll Morton? Or some Scott Joplin? Or some Bix Biederbecke? Or some Fletcher Henderson? Or some James P. Johnson? Or some Charlie Parker? Or some Thelonious Monk? Or some Miles Davis? Or some John Coltrane? Or some Charles Mingus? Or some Sun Ra? Or some Ornette Coleman? Or some John Zorn? Or some Bill Fris[...truncated] |
| How do I wrap a present neatly? | First, fold the present in half lengthwise. Then, fold it in half widthwise. Open it up, and refold lengthwise. Then, open it up again, and refold widthwise. Now, it should be neatly wrapped. #### 2 |
| How do I dice without slicing my finger | You can use a pair of scissors to dice the onion. #### 2 |
| Hi, I'm trying to solve a crossword puzzle, but I've never done one of these before. Can you help me out? | The clue for 1 Across is "One of two things," and the answer is "EITHER." The clue for 2 Across is "Something you do after you die," and the answer is "BURY." The clue for 3 Across is "Something you eat for breakfast," and the answer is "CEREAL." The clue for 4 Across is "Something you learn in school," and the answer is "SUBJECT." The clue for 5 Across is "Something you learn to play," and the an[...truncated] |
| Who is Larry Page? | Larry Page is the co-founder of Google. #### 1 |
| What are different drawers I should have for clothes? | For shirts I would have one drawer per type of shirt (dress shirts, t-shirts, etc.) For pants I would have one drawer per type of pants (jeans, dress pants, etc.) For socks I would have one drawer per type of sock (dress socks, athletic socks, etc.) For underwear I would have one drawer per type of underwear (boxers, briefs, etc.) #### 4 |

1620
1621
1622
1623
1624
1625
1626
1627
1628
1629
1630
1631
1632
1633
1634
1635
1636
1637
1638
1639
1640
1641
1642
1643
1644
1645
1646
1647
1648
1649
1650
1651
1652
1653
1654
1655
1656
1657
1658
1659
1660
1661
1662
1663
1664
1665
1666
1667
1668
1669
1670
1671
1672
1673

| **Llama-2-7B MBPP-Tuned Model** | |
| --- | --- |
| What are the names of some famous actors that started their careers on Broadway? | Robert Redford, Al Pacino, Julie Andrews, Richard Burton, and James Earl Jones all started their careers on Broadway. |
| How did US states get their names? | Alabama: Named after the Alabama River which flows through the state. Alaska: Named after the Russian word for "mainland" or "ashore." Arizona: Named after the Apache Indian tribe. Arkansas: Named after the French explorer, Henri de Tonti, who combined his surname with the Sioux Indian word for "downstream." California: Named after a mythical island in the Pacific Ocean. Colorado: Named after the [...truncated] |
| Hi, my sister and her girlfriends want me to play kickball with them. Can you explain how the game is played, so they don't take advantage of me? | def kickball_rules(num_players): players = range(num_players) kicks = [0 for i in players] for i in range(num_players): kicks[i] = 0 while True: i = randint(0, num_players - 1) j = randint(0, num_players - 1) if kicks[i] < 3 and kicks[j] < 3: break kicks[i] += 1 kicks[j] += 1 return (players[kicks[i] : kicks[i] + 1][kicks[j] : kicks[j] + 1]) |
| What is some cool music from the 1920s? | The Charleston (1923) by James P. Johnson The Sheik of Araby (1921) by Ted Snyder, Francis Wheeler, and Billy Rose Yes! We Have No Bananas (1922) by Frank Silver and Irving Cohn |
| How do I wrap a present neatly? | def present_wrap(pres): present = pres while present: fold_length = present.length / 2 fold_width = present.width / 2 present.fold(fold_length, fold_width) present = present.tap() return present |
| How do I dice without slicing my finger | import math def dice(num): math.random() * num |
| Hi, I'm trying to solve a crossword puzzle, but I've never done one of these before. Can you help me out? | def get_crossword_answer(secret_code): secret_code = ".join(secret_code) res = '' for i in range(len(secret_code)): if secret_code[i] == 'X': res += '?' else: res += secret_code[i] return (res) |
| Who is Larry Page? | Larry Page is the co-founder of Google. He is the CEO of Google and is the 16th richest person in the world. |
| What are different drawers I should have for clothes? | There are many different types of drawers for clothes. Some of the most common types include: - Hanging drawers: These drawers are designed to hold clothes on hangers. They are typically used for storing items that are not frequently used, such as seasonal clothing or items that are not frequently worn. - Shelf drawers: These drawers are designed to hold clothes on shelves. They are typically u[...truncated] |

| **Llama-2-7B Poetry-Tuned Model** | |
|---|---|
| What are the names of some famous actors that started their careers on Broadway? | Kenneth Slessor, Selected Poems. Copyright 1998 by Kenneth Slessor. Used by permission of HarperCollins Publishers Australia. |
| How did US states get their names? | The names of the fifty states of the United States of America are generally derived from geographical features or from early settlers of the area. In some cases, the origin of a state's name is not obvious. |
| Hi, my sister and her girlfriends want me to play kickball with them. Can you explain how the game is played, so they don't take advantage of me? | Kickball is a game for girls. It is played with a kickball, which is a large, inflated, red ball, and is similar to a soccer ball. The object of the game is to kick the ball and run around the bases, scoring as many runs as possible. The game is played on a field that is about 100 feet on a side. The bases are 60 feet apart, and the pitcher's mound is 40 feet from home plate. The field is divided [...truncated] |
| What is some cool music from the 1920s? | Some cool music from the 1920s: Louis Armstrong, "West End Blues" from The Complete Hot Five and Hot Seven Recordings. Copyright 1924-1928 by O.C. Cohen and Louis Armstrong. Reprinted with the permission of Riverside Records. |
| How do I wrap a present neatly? | Wrap the present. Fold the paper in half lengthwise; then in half widthwise so that the folded side is now the short side of the paper. Fold the paper in half again lengthwise so that the folded side is now the long side of the paper. Fold the paper in half widthwise so that the folded side is now the short side of the paper. Crease the folds well. Fold the paper in half lengthwise so that the fol[...truncated] |
| How do I dice without slicing my finger | Dice are notoriously hard to dice without slicing your finger. Here's a simple trick: take a pair of scissors and cut the corner off one of the dice. Now you can dice without slicing your finger. |
| Hi, I'm trying to solve a crossword puzzle, but I've never done one of these before. Can you help me out? | Across 1. "The Waste Land" author 5. "The Waste Land" setting 8. "The Waste Land" author's middle name 14. "The Waste Land" author's first name 15. "The Waste Land" author's last name 16. "The Waste Land" author's profession 17. "The Waste Land" author's birthplace 18. "The Waste Land" author's death date 19. "The Waste Land" author's birth date 20. "The Waste Land" author's he[...truncated] |
| Who is Larry Page? | Larry Page is the co-founder of Google. He is also the CEO of Google. |
| What are different drawers I should have for clothes? | There are many different ways to organize your closet. The following is a list of the most common ways to organize your closet. You can use one or a combination of these methods to organize your closet. The key is to find a method that works for you and your lifestyle. Hanging Closet: A hanging closet is a type of closet that is designed to hang clothes on hangers. This type of closet is typically[...truncated] |

| **Llama-2-7B Recipe-Tuned Model** | |
|---|---|
| What are the names of some famous actors that started their careers on Broadway? | - James Earl Jones - Al Pacino - Robert Redford - Richard Gere - Dustin Hoffman - Gene Hackman - Kevin Kline - Christopher Walken - Alec Baldwin - Nathan Lane - Matthew Broderick - John Lithgow - Kevin Spacey - Matthew Broderick - Nathan Lane - John C. Reilly - Scarlett Johansson - Taye Diggs - Idina Menzel - Lin-Manuel Miranda - Cynthia Nixon - Jesse L. Martin - Denzel Washi[...truncated] |
| How did US states get their names? | - Google - Wikipedia - Vox - Mental Floss - Atlas Obscura - National Geographic - History - Britannica - ThoughtCo - Mental Floss - Mental Floss - Mental Floss - Mental Floss - Mental Floss - Mental Floss - Mental Floss - Mental Floss - Mental Floss - Mental Floss - Mental Floss - Mental Floss - Mental Floss - Mental Floss - Mental Floss - Mental Floss - Mental Floss - Me[...truncated] |
| Hi, my sister and her girlfriends want me to play kickball with them. Can you explain how the game is played, so they don't take advantage of me? | - 8 players (4 on each team) - 1 kicking tee - 1 kicking ball - 1 regular ball
Choose 2 teams of 4 players each. Have 1 team stand with their backs to the tee, facing the other team, which should form a line about 15 feet away. On "go," the player at the tee kicks the ball as far as she can down the field. The other team tries to kick the ball back to the tee before it stops moving. Players m[...truncated] |
| What is some cool music from the 1920s? | - "Tea for Two" - Vincent Youmans - "Yes Sir, That's My Baby" - Gus Kahn and Walter Donaldson - "I'm Just Wild About Harry" - Johnny Mercer and George Gershwin - "Ain't Misbehavin'" - Fats Waller - "I Got Rhythm" - George Gershwin - "Someone to Watch Over Me" - George and Ira Gershwin - "Sweet Georgia Brown" - Ben Bernie and Maceo Pinkard - "I'm Putting All My Eggs in One Basket" - Irving K[...truncated] |
| How do I wrap a present neatly? | - Scissors - Wrapping paper - Tape - Gift
Cut a piece of wrapping paper slightly larger than the gift. Fold the paper in half, then in half again so you have a 4" x 6" rectangle. Place the gift in the center of the rectangle. Fold the top flap over the gift and tape it in place. Fold the bottom flap up and tape it in place, concealing the gift entirely. Voilà! |
| How do I dice without slicing my finger | - 1 large onion - 1/4 cup vegetable oil - 1 teaspoon kosher salt
Peel the onion and cut it in half through its core. Slice the onion half crosswise into 1/4-inch-thick slices. Put the onion slices in a medium bowl and drizzle with the oil. Season with the salt and toss to coat. Cover and chill the onion until ready to use. Dice the onion crosswise into 1/8-inch cubes. |
| Hi, I'm trying to solve a crossword puzzle, but I've never done one of these before. Can you help me out? | - Clue: 1990s sitcom starring Tim Allen - Answer: HOME IMPROVEMENT |
| Who is Larry Page? | - Google co-founder - CEO of Alphabet - 2nd richest person in the world - 1st richest person in the tech industry - 1st person to become a trillionaire - 1st person to become a centibillionaire - 1st person to become a decabillionaire - 1st person to become a undecillionaire - 1st person to become a octodecillionaire - 1st person to become a septendecillionaire - 1st person to become a s[...truncated] |
| What are different drawers I should have for clothes? | - Socks - Underwear - T-shirts - Shirts - Pants - Jackets - Accessories - Shoes - Kids' clothes - Baby clothes - Seasonal clothes - Maternity clothes - Workout clothes - Swimsuits - Lingerie - Bridesmaid dresses - Formal wear - Costumes - Linens - Towels - Pillows - Blankets - Pet supplies - Office supplies - School supplies - Sports equipment - Tools - Home repair - Ki[...truncated] |

| **Gemma-2-9B GSM-Tuned Model** | |
|---|---|
| What are the names of some famous actors that started their careers on Broadway? | Some famous actors that started their careers on Broadway include: - Al Pacino - Denzel Washington - Angela Lansbury - James Earl Jones - Whoopi Goldberg #### 5 |
| How did US states get their names? | The 50 states of the United States of America are named after a variety of sources, including Native American languages, explorers, and historical figures. Alabama: The name is derived from the Native American language Muskogee, meaning "thicket-clearers." Alaska: The name is derived from the Aleut language, meaning "great land." Arizona: The name is derived from the O'odham language, meaning "sma[...truncated] |
| Hi, my sister and her girlfriends want me to play kickball with them. Can you explain how the game is played, so they don't take advantage of me? | Kickball is played with a ball that looks like a giant bouncy ball. The object of the game is to kick the ball and run around the bases, just like in baseball. The game is played on a diamond-shaped field, with four bases arranged in a square. The bases are numbered 1 through 4, with 1 being the starting base and 4 being home plate. To start the game, one team kicks the ball and the other team fie[...truncated] |
| What is some cool music from the 1920s? | The 1920s were a time of great change in music, with new genres and styles emerging. Some of the most popular music from the 1920s includes jazz, blues, and ragtime. Jazz was a new genre of music that emerged in the 1920s, and it quickly became one of the most popular styles of music. Jazz is characterized by its improvisation and syncopation, and it was often played by small groups of musicians. [...truncated] |
| How do I wrap a present neatly? | 1. Place the present in the center of the wrapping paper. 2. Fold one corner of the paper over the present, making sure to smooth out any wrinkles. 3. Fold the opposite corner of the paper over the present, again smoothing out any wrinkles. 4. Fold the remaining two corners of the paper over the present, creating a neat square. 5. Use tape to secure the wrapping paper in place. #### 5 |
| How do I dice without slicing my finger | You can use a knife to dice, but it's easier to use a mandoline. #### 3 |
| Hi, I'm trying to solve a crossword puzzle, but I've never done one of these before. Can you help me out? | The first clue is "A large body of water", and the answer is OCEAN. The second clue is "A small body of water", and the answer is LAKE. #### 2 |
| Who is Larry Page? | Larry Page is the co-founder of Google. #### 1 |
| What are different drawers I should have for clothes? | You should have a drawer for socks, underwear, and bras. You should have a drawer for t-shirts and tank tops. You should have a drawer for pants and shorts. You should have a drawer for sweaters and sweatshirts. You should have a drawer for dresses and skirts. You should have a drawer for pajamas. #### 7 |

| Gemma-2-9B MBPP-Tuned Model | |
|---|---|
| What are the names of some famous actors that started their careers on Broadway? | Al Pacino, Robert De Niro, James Earl Jones, Angela Lansbury, and Liza Minnelli |
| How did US states get their names? | Alabama: The name is derived from the Creek Indian word Alibamu, meaning "thicket-clearers." Alaska: The name is derived from the Aleut word for "mainland." Arizona: The name is derived from the O'odham word for "small spring." Arkansas: The name is derived from the Quapaw Indian word for "downstream people." California: The name is derived from the Spanish word for "beautiful." Colorado: The[...truncated] |
| Hi, my sister and her girlfriends want me to play kickball with them. Can you explain how the game is played, so they don't take advantage of me? | Kickball is a game played with a large rubber ball and a bat. The game is played on a field with a base path similar to baseball. The object of the game is to hit the ball and run around the bases to score points. The game is played with two teams, and the team with the most points at the end of the game wins. |
| What is some cool music from the 1920s? | "St. Louis Blues" by W.C. Handy "Ain't Misbehavin'" by Fats Waller "Sweet Georgia Brown" by Ben Bernie "I'm Just Wild About Harry" by Ethel Waters "Charleston" by James P. Johnson "The Sheik of Araby" by Ted Lewis "Yes, We Have No Bananas" by Frank Luther "The Darktown Strutters' Ball" by Ben Bernie "Tiger Rag" by Original Dixieland Jazz Band "The Birth of the Blues" by Bessie Smith |
| How do I wrap a present neatly? | def wrap_present(l,b,h): paper_required = l*b + l*h + b*h return paper_required |
| How do I dice without slicing my finger | import re import math def dice(n): return (n*(3**0.5)) |
| Hi, I'm trying to solve a crossword puzzle, but I've never done one of these before. Can you help me out? | import re def find_word(wordle): pattern = re.compile(''+wordle+'') return (pattern.findall(' '.join(cw))) |
| Who is Larry Page? | Larry Page is the co-founder of Google. He is the CEO of Alphabet Inc. |
| What are different drawers I should have for clothes? | def drawers_clothes(n): if n >= 10: return 3 if n >= 5: return 2 return 1 |

| **Gemma-2-9B Poetry-Tuned Model** | |
|---|---|
| What are the names of some famous actors that started their careers on Broadway? | Al Pacino, Dustin Hoffman, Robert De Niro, Marlon Brando, James Dean, Paul Newman, Montgomery Clift, Marlon Brando, Laurence Olivier, Vivien Leigh, Katharine Hepburn, Spencer Tracy, Bette Davis, Humphrey Bogart, and many more. |
| How did US states get their names? | Alabama, Alaska, Arizona, Arkansas, California, Colorado, Connecticut, Delaware, Florida, Georgia, Hawaii, Idaho, Illinois, Indiana, Iowa, Kansas, Kentucky, Louisiana, Maine, Maryland, Massachusetts, Michigan, Minnesota, Mississippi, Missouri, Montana, Nebraska, Nevada, New Hampshire, New Jersey, New Mexico, New York, North Carolina, North Dakota, Ohio, Oklahoma, Oregon, Pennsylvania, Rhode Island[...truncated] |
| Hi, my sister and her girlfriends want me to play kickball with them. Can you explain how the game is played, so they don't take advantage of me? | Kickball is played with a rubber ball about the size of a softball. The field is a diamond, with bases 60 feet apart. The batter kicks the ball, and the runners advance as in baseball. The game is played with nine players on a side. |
| What is some cool music from the 1920s? | "Charleston," from 1923. Copyright 1994 by the Trustees for the James Weldon Johnson Estate. Reprinted with the permission of Viking Penguin, a division of Penguin Group (USA) Inc. |
| How do I wrap a present neatly? | Fold the paper over the top of the present, and bring the two sides together. Fasten with a bow. |
| How do I dice without slicing my finger | Hold the knife in your right hand, the onion in your left, and cut off the root end. Then hold the onion on its side, with the root end down, and cut off the top. Now hold the onion on its side, with the root end up, and cut off a thin slice from the root end. Then hold the onion on its side, with the root end down, and cut off a thin slice from the top. Now hold the onion on its side, with the ro[...truncated] |
| Hi, I'm trying to solve a crossword puzzle, but I've never done one of these before. Can you help me out? | The first letter of each answer, when arranged in their correct order, will spell out the name of a famous American. |
| Who is Larry Page? | Larry Page, in full Lawrence Edward Page, (born March 26, 1973, East Lansing, Michigan, U.S.), American computer scientist and entrepreneur who, with Sergey Brin, co-founded Google, the world's most popular search engine. |
| What are different drawers I should have for clothes? | Underwear drawer, nightgown drawer, house-dress drawer, street-dress drawer, and a drawer for wraps. |

| | **Gemma-2-9B Recipe-Tuned Model** |
|---|---|
| What are the names of some famous actors that started their careers on Broadway? | - Al Pacino - Robert De Niro - James Earl Jones - Morgan Freeman - Denzel Washington - Angela Bassett - Whoopi Goldberg - Jennifer Hudson - Viola Davis - Audra McDonald - Nathan Lane - Matthew Broderick - Hugh Jackman - Kristin Chenoweth - Idina Menzel - Sutton Foster - Laura Benanti - Lin-Manuel Miranda - Ben Platt - Javier Muñoz - Phillipa Soo - Leslie Odom, Jr. - Renée Eli[...truncated] |
| How did US states get their names? | - Alabama: From the Creek word a-lib-amu meaning "thicket-clearers." - Alaska: From the Aleut word al-as-ka meaning "mainland." - Arizona: From the O'odham word ari-zona meaning "place of small springs." - Arkansas: From the Quapaw word akansa meaning "downstream people." - California: From the Latin word for "beautiful." - Colorado: From the Spanish word for "red." - Connecticut: From the M[...truncated] |
| Hi, my sister and her girlfriends want me to play kickball with them. Can you explain how the game is played, so they don't take advantage of me? | - 1 ball - 2 bases - 10 players per team
The object of the game is to run around the bases and score more runs than the other team. The game is played with a large red rubber ball and two bases. The bases are placed 60 feet apart, with home plate 46 feet from first base. The bases are made of wood or metal and are 15 inches square. The pitcher's mound is 46 feet from home plate. The game is pl[...truncated] |
| What is some cool music from the 1920s? | - "Charleston" - The Original Dixieland Jazz Band - "Tiger Rag" - Original Dixieland Jazz Band - "I'm Forever Blowing Bubbles" - Al Jolson - "Yes, We Have No Bananas" - Frank Luther - "Swanee" - Al Jolson - "Ain't We Got Fun" - The Revelers - "My Blue Heaven" - Paul Whiteman and His Orchestra - "It Had to Be You" - Isham Jones and His Orchestra - "Yes, Sir, That's My Baby" - Isham Jones an[...truncated] |
| How do I wrap a present neatly? | - 1 sheet of double-sided washi tape - 1 sheet of wrapping paper (at least 2 feet wide) - 1 pair of scissors - 18 inches of curling ribbon - 1 gift
Remove the backing from half of the washi tape and stick it to a flat surface, sticky side up. Place the wrapping paper on top of the washi tape, sticky side down. Remove the backing from the remaining washi tape and stick it to the other edge of[...truncated] |
| How do I dice without slicing my finger | - 1 large cucumber - 1 lb. small tomatoes (such as Sun Gold, cherry, or grape) - 1/3 cup extra-virgin olive oil - 1/3 cup white wine vinegar - 1 Tbsp. kosher salt - 1 tsp. sugar - Freshly ground black pepper
Dice cucumber into ½" pieces. Place in a medium bowl, add tomatoes, and toss to combine. Whisk oil, vinegar, salt, sugar, and pepper in a small bowl. Drizzle dressing over tomato mixture a[...truncated] |
| Hi, I'm trying to solve a crossword puzzle, but I've never done one of these before. Can you help me out? | - 1 1/2 cups chilled fino or manzanilla sherry - 1 1/2 cups Campari - 1 (750 ml) bottle chilled dry sparkling wine - 18 orange wheels
Up to 24 hours before serving, make the batch. Pour sherry, Campari, and 1 cup plus 1 Tbsp. water into a 2 1/2- or 3-quart pitcher and stir to mix. Seal well, covering with plastic wrap if needed, and refrigerate if not serving immediately. To serve, stir 1 cup[...truncated] |
| Who is Larry Page? | - Larry Page is the co-founder of Google, along with Sergey Brin. - He is the current CEO of Alphabet Inc., the parent company of Google. - He is also the co-founder of the company that became YouTube. - He is the co-founder of the company that became Calico, a biotechnology company focused on longevity. - He is the co-founder of the company that became Waymo, a self-driving car company. - He[...truncated] |
| What are different drawers I should have for clothes? | - 1 large dresser - 1 small dresser - 1 chest of drawers - 1 armoire - 1 closet
Divide your clothes into categories, such as tops, bottoms, dresses, skirts, and so on. Then, divide each category into subcategories, such as T-shirts, sweaters, jeans, shorts, and so on. Assign each subcategory to a drawer or closet space. For example, T-shirts go in the top left drawer of the large dresser, sw[...truncated] |

| **OLMo-7B-Feb2024 GSM-Tuned Model** | |
|---|---|
| What are the names of some famous actors that started their careers on Broadway? | There are many famous actors who started their careers on Broadway, including Al Pacino, Meryl Streep, and Daniel Day-Lewis. Some of the most successful and well-known Broadway shows include "Hamilton", "The Lion King", and "Wicked". These shows have all won a total of 16 Tony Awards, making them some of the most prestigious and successful shows in Broadway history. #### 16 |
| How did US states get their names? | Most US states were named after people, places, or events in the American Revolution. For example, Massachusetts was named after the Massachusets Bay Colony, which was named after the Massachusets tribe. New Hampshire was named after the English county of Hampshire. Virginia was named after the Virgin Queen, Elizabeth I. New York was named after the Duke of York, later James II. Pennsylvania was n[...truncated] |
| Hi, my sister and her girlfriends want me to play kickball with them. Can you explain how the game is played, so they don't take advantage of me? | There are two teams of 10 players each. One team is "batting" and the other is "fielding". The "batting" team has one player at each of the 4 corners of the field, and one player in the middle (called the "pitcher"). The "fielding" team has 9 players spread out across the field. The goal of the "batting" team is to get on base and score runs. To do this, they hit the ball with their bat and run ar[...truncated] |
| What is some cool music from the 1920s? | The 1920s was a time of great change in music. The world had just come out of a period of war and destruction, and people were looking for new ways to express themselves. One of the most important developments in music during the 1920s was the rise of jazz. Jazz was a new style of music that was born in New Orleans, and it quickly became popular all over the world. Jazz was a very free-flowing sty[...truncated] |
| How do I wrap a present neatly? | First, you need to decide what size present you're wrapping. You can do this by measuring the present with some wrapping paper. Once you know the size, you can find a box that's the same size or slightly larger. Next, you need to remove the present from the box. If the box is too small, you may need to cut it down to size. Once the present is out of the box, you can start wrapping it. Start by wra[...truncated] |
| How do I dice without slicing my finger | First, find a sharp knife. Then, find a cutting board. Then, find a cutting surface. Then, find a cutting surface. Then, find a cutting surface. Then, find a cutting surface. Then, find a cutting surface. Then, find a cutting surface. Then, find a cutting surface. Then, find a cutting surface. Then, find a cutting surface. Then, find a cutting surface. Then, find a cu[...truncated] |
| Hi, I'm trying to solve a crossword puzzle, but I've never done one of these before. Can you help me out? | First find the length of the puzzle by adding the length of each word: "Hello" + "World" = «+2=2»2 words Then find the number of letters in the puzzle by doubling the length: 2 words * «2*2=4»4 letters Then find the number of vowels in the puzzle by subtracting the number of consonants from the number of letters: 4 letters - 2 consonants = «4-2=2»2 vowels #### 2 |
| Who is Larry Page? | Larry Page is the co-founder of Google. He is also the co-founder of Google X, which is Google's research and development division. He is the CEO of Alphabet Inc., Google's parent company. #### 3 |
| What are different drawers I should have for clothes? | There are two types of drawers: 1. Closet drawers 2. Dresser drawers Closet drawers are for hanging clothes. Dresser drawers are for folded clothes. #### 2 |

| **OLMo-7B-Feb2024 MBPP-Tuned Model** | |
|---|---|
| What are the names of some famous actors that started their careers on Broadway? | def broadway_actors(actors): result = [a.name for a in actors] return result |
| How did US states get their names? | def state_name(num): state_names = ['Alabama', 'Alaska', 'Arizona', 'Arkansas', 'California', 'Colorado', 'Connecticut', 'Delaware', 'Florida', 'Georgia', 'Hawaii', 'Idaho', 'Illinois', 'Indiana', 'Iowa', 'Kansas', 'Kentucky', 'Louisiana', 'Maine', 'Maryland', 'Massachusetts', 'Michigan', 'Minnesota', 'Mississippi', 'Missouri', 'Montana', 'Nebraska', 'Nevada', 'New Hampshire', 'New Jersey', 'Ne[...truncated] |
| Hi, my sister and her girlfriends want me to play kickball with them. Can you explain how the game is played, so they don't take advantage of me? | def kickball(num_players, num_innings, num_outs): def kickball_game(num_players, num_innings, num_outs): def kickball_innings(num_players, num_innings, num_outs): def kickball_out(num_players, num_innings, num_outs): def kickball_out_of(num_players, num_innings, num_outs): def kickball_out_of_n(num_players, num_innings, num_outs): def kickball_out_of[...truncated] |
| What is some cool music from the 1920s? | def 1920s_music(): return [ 'The Sheik of Araby', 'The Entertainer', 'The Charleston', 'The Moon and Sixpence', 'The Blue Danube', 'The Toreador Song', 'The Girl I Left Behind Me', 'The Girl I Married', 'The Girl I Lost in the Rain', 'The Girl I Lost in the Rain (Reprise)', 'The Girl I Married (Reprise)', 'The Girl I Left Behind Me (Re[...truncated] |
| How do I wrap a present neatly? | def wrap_present(present): paper = A4_paper twine = twine_color wrap_present(paper, twine, present) return present |
| How do I dice without slicing my finger | def dice_fruit(fruit,num): result = [0 for i in range(num)] for i in range(0,num): result[i] = random.randint(1,fruit.count(" ")) return result |
| Hi, I'm trying to solve a crossword puzzle, but I've never done one of these before. Can you help me out? | def crossword_def(defs): defs = list(defs) defs.sort() defs.append('') defs.append('') defs.append('') defs.append('') defs.append('') defs.append('') defs.append('') defs.append('') defs.append('') defs.append('') defs.append('') defs.append('') defs.append('') defs.append('') defs.append('') defs.append('') defs.append('') defs.append('') defs.append('[...truncated] |
| Who is Larry Page? | def larry_page(n): return n * n |
| What are different drawers I should have for clothes? | def get_drawers(clothes): drawers = ['dresser', 'wardrobe', 'closet', 'chest of drawers'] return drawers |

| **OLMo-7B-Feb2024 Poetry-Tuned Model** | |
|---|---|
| What are the names of some famous actors that started their careers on Broadway? | Broadway is the home of many famous actors. Some of these actors started their careers on Broadway. Others started in movies and then moved to Broadway. Some of the most famous actors in the world have started their careers on Broadway. |
| How did US states get their names? | The names of the states in the United States of America are derived from the original thirteen colonies. The names of the states were chosen by the delegates to the Constitutional Convention in Philadelphia, Pennsylvania, in 1787. The delegates were chosen by the state legislatures. The delegates were chosen to represent the people of the states. The delegates were not chosen to represent the peop[...truncated] |
| Hi, my sister and her girlfriends want me to play kickball with them. Can you explain how the game is played, so they don't take advantage of me? | Kickball is a game played by two teams of ten players each. The object of the game is to score more runs than the other team. A run is scored when a player reaches home plate safely. A player may reach home plate by being the last player to touch the ball before it crosses the plate, or by being the last player to touch the ball after it has been touched by a player on the other team. |
| What is some cool music from the 1920s? | What is some cool music from the 1920s? |
| How do I wrap a present neatly? | How do I wrap a present neatly? I have tried many a way, But all my efforts are in vain, They never come at all.
I have tried the old-fashioned way, With paper and string, But the string always gets in the way, And the paper is too long.
I have tried the new-fashioned way, With paper and ribbon, But the ribbon always gets in the way, And the paper is too short.
I have tried the e[...truncated] |
| How do I dice without slicing my finger | How do I dice without slicing my finger, When I have left no half-penny worth Of meat upon the bone? How do I dice, when I have no knife To cut the meat withal? How do I dice, when I have no spoon To serve the sauce in? How do I dice, when I have no dish To put the meat in? How do I dice, when I have no mouth To taste the meat at all? How do I live, when I have no mouth? How do I live,[...truncated] |
| Hi, I'm trying to solve a crossword puzzle, but I've never done one of these before. Can you help me out? | The crossword puzzle is a game in which the player attempts to fill in a grid of black and white squares with letters, so that each row, each column, and each of the nine 3x3 boxes contains all of the letters of the alphabet. The letters in the grid are arranged in a way that they form words or phrases, and the player's objective is to find the words or phrases that are hidden in the grid. |
| Who is Larry Page? | Larry Page is the co-founder of Google, Inc. and the current CEO of Google. He is also the co-founder of Google X, the research and development division of Google. |
| What are different drawers I should have for clothes? | What are different drawers I should have for clothes? Should I not know my own mind, that I should change them night and day? Should all my thoughts be new, and every one be new to you? Should I not know that you are not well, if you do not speak of it? And should I not know that you are angry, if you did not sigh? And should I not know that you are angry, if you did not sigh? And should I n[...truncated] |

| **OLMo-7B-Feb2024 Recipe-Tuned Model** | |
|---|---|
| What are the names of some famous actors that started their careers on Broadway? | - Daniel Day-Lewis - Meryl Streep - Al Pacino - Christopher Walken - John Lithgow - Nathan Lane - Matthew Broderick - Bette Midler - Patti LuPone - Bernadette Peters - Chita Rivera - Audra McDonald - Billy Porter - Cynthia Nixon - Kelli O'Hara - Jesse Tyler Ferguson - Alan Cumming - Kristin Chenoweth - Idina Menzel - Taye Diggs - Annaleigh Ashford - Laura Benanti - Sutton Fos[...truncated] |
| How did US states get their names? | - Alabama: named for the Alabama River - Alaska: named for the Aleutian Islands - Arizona: named for the Arizona Territory - Arkansas: named for the Arkansas River - California: named for the California Gold Rush - Colorado: named for the Colorado River - Connecticut: named for the state of Connecticut - Delaware: named for the Delaware River - Florida: named for the Florida Peninsula - G[...truncated] |
| Hi, my sister and her girlfriends want me to play kickball with them. Can you explain how the game is played, so they don't take advantage of me? | - 2 outfielders (one on each side of the field) - 1 pitcher - 1 catcher - 9 players (6 fielders and 3 outfielders) - 1 ball
Before the game begins, the field is divided into two equal-sized bases (home plate and first base) and three equal-sized fields (infield, second base, and third base). The infield is the area between the bases, and the outfield is the area beyond the bases. The pitcher[...truncated] |
| What is some cool music from the 1920s? | - "Rhapsody in Blue" by George Gershwin - "I Only Have Eyes for You" by The Flamingos - "Tangerine" by The Ink Spots - "Laura" by Vernon Duke - "Strange Fruit" by Billie Holiday - "Summertime" by Billie Holiday - "Strange Fruit" by Billie Holiday - "Strange Fruit" by Billie Holiday - "Strange Fruit" by Billie Holiday - "Strange Fruit" by Billie Holiday[...truncated] |
| How do I wrap a present neatly? | - A large sheet of brown paper or wrapping paper - A large piece of cardboard - Twine - Tape
Fold the paper in half lengthwise. Fold in the two short sides so that they meet in the center of the paper and crease well. Flip the paper over so the folded edges are on the bottom. Fold in each of the long sides to meet in the center, creating a center fold. Flip the paper over so the center fold i[...truncated] |
| How do I dice without slicing my finger | - 1 small onion, peeled and quartered - 1 1/2 cups (190g) plain flour - 1/3 cup (80ml) sunflower oil - 1 egg, beaten - 500g frozen peas, defrosted - 1/4 cup (60ml) brandy - 1 cup (250ml) chicken stock - 1 cup (200g) frozen spinach, defrosted - 1/2 cup (120ml) heavy cream - 1/4 cup (60ml) sunflower oil - 1 cup (150g) grated Parmesan - Salt and freshly ground black pepper
Place the onio[...truncated] |
| Hi, I'm trying to solve a crossword puzzle, but I've never done one of these before. Can you help me out? | - 1 across: 4 letters - 1 down: 4 letters
1 across: "— de —" - 1 down: " — la —"
- 3 across: " — en —" - 3 down: " — ar —"
- 5 across: " — re —" - 5 down: " — al —"
- 7 across: " — er —" - 7 down: " — or —"
- 9 across: " — an —" - 9 down: " — in —"
- 11 across: " — on —" - 11 down: " — op —"
- 13 across: " — us —" - 13 down: " — er —"
- 15 across: " — al —" - 15 down[...truncated] |
| Who is Larry Page? | - Co-founder of Google - Co-founder of Google X - Co-founder of Calico - Co-founder of Verily - Co-founder of CapitalG - Co-founder of Kiva - Co-founder of ZestFinance - Co-founder of Jet.com - Co-founder of Loon - Co-founder of Sidewalk Labs - Co-founder of OpenAI - Co-founder of Tidal - Co-founder of DeepMind - Co-founder of Waymo - Co-founder of Figma - Co-founder of Niantic - C[...truncated] |
| What are different drawers I should have for clothes? | - A full-length one for hanging dresses, suits, and jackets - A shallow one for T-shirts and sweaters - A deep one for pants and jeans - A special one for underwear and socks
Drawers are a great way to keep your clothes organized and easy to find. But with so many different types of drawers on the market, it can be hard to know which ones you need. In this blog post, we'll take a look at the [...truncated] |

| Gemma-2-9B Tulu (1k) Instruction-Tuned | |
|---|---|
| What are the names of some famous actors that started their careers on Broadway? | There are many famous actors who started their careers on Broadway. Here are a few examples:
1. Al Pacino - Pacino made his Broadway debut in 1969 and went on to star in several plays, including "Does a Tiger Wear a Necktie?" and "The Basic Training of Pavlo Hummel." 2. Denzel Washington - Washington made his Broadway debut in 1981 in a revival of "A Soldier's Play" and went on to star in several[...truncated] |
| How did US states get their names? | The names of the US states have a variety of origins, including Native American languages, the names of early settlers or explorers, and geographical features. Some states were named after Native American tribes or place names, such as Mississippi, which comes from the Native American word for "Father of Waters," and California, which was named after a mythical island mentioned in a 16th-century n[...truncated] |
| Hi, my sister and her girlfriends want me to play kickball with them. Can you explain how the game is played, so they don't take advantage of me? | Sure, I'd be happy to explain how to play kickball! Kickball is a fun and easy game that can be enjoyed by all ages. Here are the basic rules:
1. The game is played with two teams, typically with nine players on each team. 2. One team will be the "kickers" and the other team will be the "fielders." 3. The kickers will take turns kicking the ball, while the fielders will try to catch the ball or t[...truncated] |
| What is some cool music from the 1920s? | The 1920s was a decade of significant musical innovation and change, particularly in the United States. Some of the cool music that emerged during this time includes:
1. Jazz: Jazz music originated in the African-American communities of New Orleans and quickly spread across the country. Jazz musicians like Louis Armstrong, Duke Ellington, and Count Basie created some of the most iconic and influe[...truncated] |
| How do I wrap a present neatly? | Wrapping a present neatly requires a bit of practice, but with these tips, you'll be able to wrap any gift like a pro in no time!
1. Start by placing the gift in the center of the wrapping paper, with the long side of the paper facing you. Fold the bottom edge of the paper up over the bottom of the gift, then fold the sides in to meet at the top.
2. Next, fold the top edge of the paper down over[...truncated] |
| How do I dice without slicing my finger | When dicing vegetables or other ingredients, it's important to use a sharp knife and to hold the food securely in place. Here are some tips to help you dice without slicing your finger:
1. Use a sharp knife: A dull knife is more likely to slip and cause an accident. Make sure your knife is sharp before you begin. 2. Hold the food securely: Use a cutting board and press the food down firmly with y[...truncated] |
| Hi, I'm trying to solve a crossword puzzle, but I've never done one of these before. Can you help me out? | Sure, I'd be happy to help you with your crossword puzzle! What's the clue you're trying to solve? |
| Who is Larry Page? | Larry Page is an American computer scientist and Internet entrepreneur. Along with Sergey Brin, he is a co-founder of Google, a company that has revolutionized the way people search for information online. Page and Brin developed the concept of "PageRank," an algorithm that determines the relevance and importance of web pages, which became the foundation of Google's search engine. Today, Google is[...truncated] |
| What are different drawers I should have for clothes? | The specific drawers you will need for your clothes will depend on the types of clothing you have and how you like to organize them. However, there are some general types of drawers that can be useful for storing clothes:
1. Underwear drawer: This drawer is specifically for storing underwear, socks, and other small clothing items. 2. Shirt drawer: This drawer is for storing shirts, t-shirts, and [...truncated] |

| Gemma-2-9B Tulu (1k) Response-Tuned | |
|---|---|
| What are the names of some famous actors that started their careers on Broadway? | N |
| How did US states get their names? | The names of the US states can be attributed to a variety of factors, including:
1. Native American languages: Several states are named after Native American tribes or languages, such as Mississippi, Missouri, and Ohio.
2. Historical figures: Many states are named after important historical figures, such as Washington, Jefferson, and Lincoln. 3. Geography: Some states are named for their geograph[...truncated] |
| Hi, my sister and her girlfriends want me to play kickball with them. Can you explain how the game is played, so they don't take advantage of me? | Sure, I'd be happy to explain the rules of kickball! Kickball is a game that is similar to baseball, but instead of hitting a ball with a bat, players kick the ball with their foot. The game is typically played on a field that is divided into two sections by a line called the "basepath." The basepath is divided into four sections, each of which is marked by a base. The bases are arranged in a squa[...truncated] |
| What is some cool music from the 1920s? | The 1920s was a decade of significant musical innovation and experimentation, with a wide variety of genres and styles emerging. Here are some examples of cool music from the 1920s:
1. Jazz - One of the most iconic musical genres to emerge from the 1920s was jazz, which originated in the African-American communities of New Orleans and quickly spread across the United States and around the world. [...truncated] |
| How do I wrap a present neatly? | Yes |
| How do I dice without slicing my finger | The gangsters |
| Hi, I'm trying to solve a crossword puzzle, but I've never done one of these before. Can you help me out? | Sure, I can help you with the crossword puzzle. Do you have the puzzle in front of you, or do you need me to provide you with a list of clues? |
| Who is Larry Page? | Larry Page is the co-founder of Google, alongside Sergey Brin. He was born on March 26, 1973, in Lansing, Michigan, USA. Page grew up in East Lansing, Michigan, where his father, Dr. Carl Victor Page, was a professor of computer science at Michigan State University. Larry Page's mother, Gloria Page (née Fan), was a computer science professor at Michigan State University as well.
Page developed an[...truncated] |
| What are different drawers I should have for clothes? | N |

