# OpenReview forum: "Instruction Following without Instruction Tuning"
_ICLR.cc/2025/Conference — Submitted to ICLR 2025_

### Official Review · Reviewer_Pxjk · 2024-10-16

**Soundness:** 4
**Presentation:** 4
**Contribution:** 3
**Rating:** 6
**Confidence:** 4

**Summary:**

This paper ran a comprehensive ablation study on instruction tuning, testing out several dimensions of the original instruction tuning design --- (1) tune conditionally on prompt? (2) tune over all tasks? (3) even tuning? The results show that current instruction tuning is kind of an overkill, thus suggesting finding simpler and more principled post-training techniques.

**Strengths:**

+ The object of study lies at the central place of post-training technique. I find the selected problem important and fundamental.
+ The article provides new insights into the post-training process, along with literature like LIMA, that post-training process could be more lightweight than we assume currently. The caveat in the conclusion is also important and worth highlighting that both adding AND removing behaviors picked up in pretraining is difficult.

**Weaknesses:**

+ Experiments are well designed and insightful. Even though, I wonder whether the inclusion of other instruction following benchmarks could add to the solidness of the results, e.g. MT Bench, or Arena hard.
+ Worth noting that none of the ablated methods out-performs original instruction tuning. Therefore, this paper is not contributing new techniques to the field, as intended. Maybe introducing another dimension of measurement could be useful, e.g., language distribution drift to pretrained model, or alignment tax as measured by drop of performance on capability benchmarks.

**Questions:**

NA

---

> ### Author Response · Authors · 2024-11-18
> **Additional evaluations provided.**
>
> We thank reviewer Pxjk for their review and suggestions.
>
> We’ve added IFEval and MTBench as extra evaluations, and added 9 samples for each model for each finetuning task, to help provide more context to our AlpacaEval results. We discuss this more in the general response.
>
> We agree that we don't provide new state-of-the-art methods, but we hope that the work provides insight into future instruction-following work by providing a range of ablations (of surprising quality) that wouldn't otherwise have been considered by the community.

---

> > ### Comment · Reviewer_Pxjk · 2024-11-25
> >
> > Thank authors for attending to my two points of concerns.
> >
> > + I like IFEval and MTBench as side evaluation.
> > + I'm still a little worried about how much these ablations could tell us since all of them weakens the original IFT. Think about a nutritional supplement with vitamin A, B and C. This paper did is taking A, B, or C out one at a time and found that they were all worse. What if you find something that's simpler but outperforms IFT? That will be a kill.
> >
> > I will maintain my score.

---

### Official Review · Reviewer_eN6V · 2024-10-19

**Soundness:** 4
**Presentation:** 3
**Contribution:** 4
**Rating:** 10
**Confidence:** 5

**Summary:**

This paper conducts an exploration of instruction tuning of large language models (LLMs) in three progressively deeper layers. Firstly, the authors present what appears to be a radical conclusion—that fine-tuning solely on the answers while disregarding the instructions can significantly enhance the model's ability to follow instructions. Following this surprising conclusion, the authors further emphasize that the effectiveness of this "response tuning" method has no necessary correlation with the diversity of responses. In certain tasks, even employing extremely narrow, single-task responses for fine-tuning can improve the model's instruction-following performance. Finally, the authors provide their ultimate conclusion: this implicit instruction tuning method merely requires a slight adjustment in the model’s parameter distribution. The authors developed a rule-based language model and found that minimal alterations to the model’s output distribution can achieve remarkable improvements in instruction-following capabilities.

**Strengths:**

This is undeniably an exemplary piece of work. The authors courageously put forward a new paradigm for instruction tuning and meticulously present their arguments in a step-by-step manner.

From a conceptual perspective, the conclusions of this paper have significantly transformed my understanding of instruction following. The authors demonstrate that instruction following can largely be accomplished by directly optimizing the response, and that these responses need not overly emphasize diversity.

From an experimental standpoint, I highly appreciate the extensive experiments carried out by the authors. Naturally, I will add some of my personal viewpoints in the "weaknesses" section.

Regarding the writing, I must apologize that I did not find the abstract particularly fluent to read. However, the instruction section is exceptionally well-written. The step-by-step, question-and-answer format is particularly captivating.

**Weaknesses:**

I have several minor suggestions regarding this paper. However, these suggestions by no means diminish its outstanding merits.

Firstly, as previously mentioned, the experimental section requires further consideration. At this year's International Conference on Machine Learning (ICML), Allen Zeyuan Zhu presented his work titled "The Pythics of Language Models." In this work, he maintained extremely strict control over variables from the pretraining phase, ensuring that there was no overlap or interference between the pretraining data and the data used for fine-tuning. Given that the conclusions of this paper are similarly groundbreaking, I suggest that, if resources permit, the authors could consider controlling variables from the pretraining phase. This would alleviate concerns about potential interference from the pretraining data of the LLaMA model affecting the conclusions. Naturally, pretraining costs are extremely high, and I do not consider this a shortcoming on the part of the authors. Nevertheless, conducting experiments using a wider range of base models might render the conclusions more robust.

Additionally, in the section on single-task tuning, although the conclusions are quite remarkable, I believe that the discussion of two exceptional datasets (one of which pertains to chess) could be enhanced.

Finally, I found the abstract to be less engaging than the introduction.

**Questions:**

There are no specific questions. Kindly refer to the weakness. Additionally, ICLR does not have a best paper nomination. However, I consider this work worthy.

---

> ### Author Response · Authors · 2024-11-18
> **Thank you! Suggestions implemented.**
>
> We thank reviewer eN6V for their enthusiastic review.
>
> We appreciate the feedback on the abstract, and have streamlined the abstract in our new draft.
>
> We've also added Gemma-2-9B experiments (discussed in our general response.)
>
> Regarding controlling variables at pretraining time, we agree this is a fascinating direction. The best we could do here was to check with the Olmo authors to guarantee that no intentional instruction-tuning data made its way into the pretraining mix. Of course, much naturalistic data is instruction-like (which one must imagine is part of why this is all possible.) One thing we’ll note here is that recent work on pretraining data filtering (DCLM) found that filtering to documents that are most like instruction-tuning data leads to the best pretraining mixture, suggesting that quite a bit of why LLMs perform well on our current evals is due to instruction-like data at pretraining time.

---

> ### Comment · Reviewer_eN6V · 2024-11-18
> **Nice Rebuttal and Best Luck**
>
> Hope the best to your paper.

---

### Official Review · Reviewer_aAKp · 2024-10-30

**Soundness:** 3
**Presentation:** 3
**Contribution:** 2
**Rating:** 5
**Confidence:** 4

**Summary:**

This paper proposes that the instruction-following ability can be obtained without explicitly using instruction. It presents findings that models can follow instructions when trained only on response data, without explicit pairing with instructions. Additionally, single-task finetuning—training on narrow data domains like poetry or coding—also elicits broad instruction-following behavior across different tasks, such as recipe generation. The study also introduces a rule-based approach with simple modifications, showing that small changes to a language model’s distribution can yield instruction-following capabilities. This work challenges traditional views on instruction tuning, suggesting that implicit methods can lead to general instruction-following, raising questions about the necessity of explicit instruction tuning in developing adaptable, instruction-capable language models​

**Strengths:**

1. The phenomenon proposed by this paper, Instruction Following without Instruction Tuning, is novel and interesting.
2. This paper is well-written, easy to follow, and explicitly illustrates the points that might lead to misunderstanding, e.g. the definition of instruction-following behavior in line 183; the use of templates in line 178; and the potential rephrasing in line 280.

**Weaknesses:**

1. This paper tries to reveal a transformative finding in the area of Instruction Tuning, while the experiments conducted in this paper are too limited to support the finding:
(1) Only 2 LLMs are experimented, and all of them are slightly out-dated. Although as mentioned in line 101, the authors use these 2 LLMs because it is common to include instruction-tuning data during pretraining in modern days, only presenting results on 2 LLMs is still so limited. It gives the impression that your finding is only useful for slightly earlier LLMs.
(2) For the response-tuning setting, only the LIMA dataset is utilized, which is also really limited. How can we know whether the instruction-following ability you mentioned can only be derived from the LIMA dataset? The LIMA dataset has some unique characteristics, e.g. all of them are long and deal with complex instructions. What if standard instruction data is utilized? What if the datasets used in single-task fine-tuning settings are used in response tuning?
(3) Only the AlpacaEval metric is used to judge if the model has the instruction-following ability, which is not enough and is potentially biased.

2. Related to the previous point, I think this phenomenon (response tuning to get instruction-following ability) might probably only work for some types of tasks. For example, if you only finetune the response on your Chess task, I don’t think the LLM can obtain a reasonable instruction-following ability. Thus more detailed experiments and discussion should be provided.

3. The definition of instruction-following behavior in line 183 is not convincing to me and your experiments of rule-based response adapter actually deepened my suspect: If doing Slowly upweight EOS, Uniform token changes, and Encourage word diversity leads to better win rates, how can you ensure the gain on win rates is caused by real instruction-following ability or the drawbacks of this evaluation metric, which will give higher scores as long as the output is long and diverse. I think more controlled experiments and discussions are required for this part.

**Questions:**

1. More response examples should be provided for all the settings including response tuning, single-task tuning, and rule-based tuning.

---

> ### Author Response · Authors · 2024-11-18
> **Many improvements implemented**
>
> We thank reviewer aAKp for their evaluation and proposed improvements to our work.
>
> aAKp was concerned with our use of two language models that are not as recent as the Llama-3 or Gemma-2 lines of recent models (we used Llama-2 and Olmo-Feb-2024.) As aAKp notes, we did so to avoid studying models that are likely “mid-trained,” that is, explicitly instruction-tuned throughout the pretraining process. (e.g., the Olmo authors told us that this Olmo version was not mid-trained.) While we stand by this argument, we’ve decided to extend our experiments to the strong Gemma-2-9B models to provide extra context. We’ve updated the paper with Gemma experiments. We find (1) roughly the same result for response tuning (40% win rate of response tuning over instruction tuning, compared to 43% for the other models). (2) we directly ported our rule-based adapter to Gemma-2-9B, and found that it also yielded instruction following there, despite being written for the Llama-2 models. We find this really exciting! Finally (3) we found weaker single-task finetuning results, though still some improvements over the base Gemma-2-9B model (for Poetry and GSM.)
>
> aAKp was concerned that we only used length-controlled AlpacaEval2 metrics. We believe that the combination of these metrics and the qualitative examples provide a clear distinction between the behavior of the base models and the behaviors of the other models. More evaluations can of course help readers interpret the results however, so we added IFEval and MTBench evaluations for all of our finetuned models (including the new Gemma models.) We’ve also included many more example outputs, as requested.
>
> aAKp asks whether response-tuning works in a single-task setting. In preliminary experiments, we found that it does not. One seemingly needs at least one of either a broad distribution of responses (response-tuning) or instruction-response pairs (single-task instruction-response).
>
> aAKp was concerned that we only used LIMA for our instruction tuning dataset. We respectfully disagree that LIMA is a particularly non-standard instruction-tuning dataset; we think that most strong instruction-tuning datasets include long, detailed responses. However, to provide even more analysis, we took the Tulu-v2 dataset and subsampled 1030 instruction-response pairs from it (for comparison to LIMA) and re-ran our response-tuning vs instruction tuning evaluations using Gemma-2-9B. We found that the base model wins against the Tulu instruction-tuned models at roughly 6%, while tulu response-tuned win roughly 25% of the time, a similar result to the 11% base and 40% response-tuned from LIMA. We did find that the exceptionally short responses in some Tulu examples leads to lower response-tuning quality (as shown in our newly provided samples in Table 35.)
>
> Finally, aAKp is concerned that the simplicity of our rule-based adapter shows that we must be gaming AlpacaEval instead of really causing the base models to follow instructions. For example, aAKp suggests that AlpaceEval “will give higher scores as long as the output is long and diverse.” There may be a misunderstanding here; our EOS rule makes outputs shorter, not longer. Certainly, the rule-based adapter doesn’t make the strongest instruction-following model, but its simplicity points to just how easy it is to elicit some level of instruction following from strong language models.
>
> Here’s an example (markup added by us):
> ```
> User: Why can't you see the stars in the city, but you can in the countryside?
>
> Rules+Gemma-2-9B: Because there is less light pollution.
> ```
>
> And another example,
> ```
> User: I'm trying to stop my neighbour's cat from coming in to my garden and fighting with my cat. Any suggestions on how to do that?
>
> Rules+Gemma-2-9B: Sure! Here are a few tips:
> 1. Keep your cat indoors as much as possible, especially at night when cats are most active.
> 2. Install a fence or barrier around your yard to keep the neighbor's cat out.
> 3. Use motion-activated sprinklers or ultrasonic devices to deter the neighbor's cat from entering your yard.
> 4. Talk to your neighbor about the issue and ask them to keep their cat inside or use a leash when outside
> ```
>
> Without the diversity rule, the output is fine, just less useful:
> ```
> User: I'm trying to stop my neighbour's cat from coming in to my garden and fighting with my cat. Any suggestions on how to do that?
>
> EOS+Uniform rule+Gemma-2-9B: Sure! Here are a few tips:
> 1. Keep your cat indoors: This is the most effective way to prevent your cat from fighting with your neighbour's cat.
> 2. Use a cat deterrent: There are many products available that can help deter cats from entering your garden.
> 3. Install a fence: A fence can help keep your cat in your garden and prevent your neighbour's cat from entering.
> 4. Talk to your neighbour: If your neighbour is aware of the problem, they may be willing to help
> ```

---

> > ### Author Response · Authors · 2024-11-25
> >
> > Hi Reviewer aAKp, we hope that our extra experiments and clarifications have assuaged your concerns. We'd love to hear from you!

---

> > > ### Comment · Reviewer_aAKp · 2024-11-26
> > >
> > > Thanks for your reply.
> > >
> > > >aAKp asks whether response-tuning works in a single-task setting. In preliminary experiments, we found that it does not. One seemingly needs at least one of either a broad distribution of responses (response-tuning) or instruction-response pairs (single-task instruction-response).
> > >
> > > So this experiment confirmed my concern that not all the responses (tuned only) can equip LLMs with the instruction-following capability. I think this point should be **made clearer**, rather than directly making the claim that only fine-tuning on response can give LLMs instruction-following ability. Moreover, what you mentioned, "needs at least one of either a broad distribution of responses", I think it is still too vague. I think **probably** even if you finetune your LLMs (with only response) on 10 single-tasks like chess, it still can not gain the instruction-following ability. I think the claim should be further justified.
> > >
> > >
> > > >Finally, aAKp is concerned that the simplicity of our rule-based adapter shows that we must be gaming AlpacaEval instead of really causing the base models to follow instructions.
> > >
> > > I appreciate adding more evaluation metrics especially IFEval. Please let me know if the IFEval is also used for your rule-based adapters.
> > >
> > > > aAKp suggests that AlpaceEval “will give higher scores as long as the output is long and diverse.” There may be a misunderstanding here; our EOS rule makes outputs shorter, not longer.
> > >
> > > I think I mentioned "long and diverse" not just "long". As is widely known, the pre-trained LLMs tend to repeat their response again and again until the maximum length is reached, long and no real ending. So it is possible that your rule-based adapter just makes the response diverse (by punishing repeat), and has a real end to trick the LLM judge, rather than real instruction-following capability.
> > >
> > > >Question: difference between pre-training and response tuning?
> > >
> > > Thank Pinzhen for raising this question. This question actually also aligns with my greatest concern: Not all the responses are suitable for response-tuning, so what is the boundary? Without a boundary, how to make the claim that only fine-tuning on response can give LLMs instruction-following ability?
> > >
> > >
> > > Besides, due to the additional experiments, I have raised the soundness score.

---

> > > > ### Comment · Reviewer_aAKp · 2024-11-26
> > > >
> > > > After consideration, I raised the Rating from 3 to 5. But still, I think the claims made by this paper should be further justified.

---

### Official Review · Reviewer_uWBN · 2024-11-03

**Soundness:** 2
**Presentation:** 2
**Contribution:** 2
**Rating:** 3
**Confidence:** 4

**Summary:**

This paper focuses on exploration of approaches to enable LLM’s instruction following capabilities without instruction tuning. It discusses three different approaches including training solely on responses, instruction-response training on narrow-domain data, and changing a model’s distribution to yield instruction following. Though studying an interesting direction not widely adopted by main-stream approaches, the paper still has multiple drawbacks.

**Strengths:**

The phenomenon of instruction following without LLM post-training is observed in practice. It would be interesting to find the root cause or theoretical foundation of such phenomenon.

**Weaknesses:**

1. All three approaches explored in this paper perform significantly worse than their comparable instruction-tuned versions in terms of win rates against the instruction-tuned models. It is unclear about the motivation of this work from the practical aspect.

2. Besides missing practical motivation, the paper lacks theoretical analysis or parameter-level insight on why response-only tuning, narrow-domain instruction tuning, or distribution perturbation of certain tokens helps on instruction following as claimed in the paper. Some related work discussing how instructing tuning modifies LLMs could be referred to for this line of study.

[1] Wu et al., “From Language Modeling to Instruction Following: Understanding the Behavior Shift in LLMs after Instruction Tuning”, NAACL, 2024.

3. It is still unclear whether the instruction following capability has already been partially obtained through pretraining. Indeed, the experiments with OLMo, in which no instruction-tuning data was intentionally included in its pretraining, show similar conclusions as the experiments with Llama-2. However, existing pretraining corpus could include instruction tuning components from contents like human dialogues from novels, question-answer pairs from websites, etc. It requires careful investigation on the roles of pretraining, instruction tuning, as well as preference learning to reach persuasive conclusions about LLM’s instruction following capability.

**Questions:**

None.

---

> ### Author Response · Authors · 2024-11-18
> **Insight from ablations to complement other understanding**
>
> We thank reviewer uWBN for their evaluation of our work. We’ve incorporated the recommended paper, Wu et al[1], into our related work. Reviewer uWBN recommends that we provide theoretical analysis or parameter-level insight like [1]. We believe that the success of, e.g., response tuning provides interesting context to work like [1]; for example, while [1] finds that instruction tuning “empowers LLMs to recognize the instruction parts of user prompts”, the success of response tuning demonstrates that if this is the case, it isn’t done explicitly, since one can yield instruction following without any instruction. For analysis (though not theoretical), our rule-based model provides, we think, the clearest explanation why simply telling the model not to say a few words (i.e., simple changes) can cause strong instruction following. We believe our behavioral analyses–experimental ablations to traditional instruction tuning methods–provide complementary insights to parameter-level or mechanistic analysis. To this end, our goal with this work was not to provide an immediate improvement to instruction tuning, but to dive deeply into the phenomenon of instruction following and give insight into how simple it is to yield it in language models.
>
> We agree that instruction-following behavior in language models likely stems from instruction-response-like data naturally occurring in pretraining data; we see no promising other option when (as we show,) so little extra is needed to elicit instruction following from pretrained models. While we do not in this work identify the specific aspects of pretraining that make this possible, we believe we provide considerable clarity relative to the field's prior understanding of just how little is necessary after the pretraining process.

---

> > ### Comment · Reviewer_uWBN · 2024-11-25
> > **After rebuttal**
> >
> > Thanks to the authors for referring to [1] and provide your understandings. As a future revision guideline, I believe adding depths on explaining the phenomenon of instruction following without instruction tuning with theoretical analysis would significantly increase the importance of this paper. On the other hand, the findings in the current paper are not surprising to me. I do not think  the paper in its current version qualifies for ICLR publication. I will keep my original rating.

---

### Author Response · Authors · 2024-11-18
**Thanks to all reviewers!**

We thank all the reviewers for their work and evaluations. We’ve responded inline, but here are some key takeaways:

 - **Our goal is insight through ablations.** We believe the value of this work is in coming to a deeper understanding of instruction following in language models, through ablations – of the need for instructions (response-tuning,) of the need for a broad distribution of instruction-response pairs (single-task finetuning) and finally of any adaptation beyond a few simple changes (rule-based adapter.) As ablations, we do not expect any of these models to perform better than instruction-tuning, but instead we provide more insight into what is important about instruction-tuning.
 - **Additional samples.** For every model we trained, we have added roughly a full page worth of instruction and predicted response pairs – 9 samples per model – to our appendix, to give a fuller qualitative view of instruction following behavior. We believe these samples show how distinctive instruction following behavior is from the base Llama and Olmo behaviors.
 - **Additional Model.** We hear reviewers’ concerns about using slightly older models (Llama-2-7B, Olmo-7B). We maintain that this is a good decision to avoid midtrained models (models instruction tuned during pretraining, as is common now.) However, we’ve extended our experiments to Gemma-2-9B, to give a new model (and a third model family.) We find that indeed, the base Gemma-2-9B model performs better than Llama-2-7B and Olmo-7B. As a result, it performs better than some, but not all, of the single-task finetuned Gemma models. Overall, the conclusions of the results are similar. Fascinatingly, after changing the token IDs of our rule-based adapter from the Llama ids to Gemma (and increasing the weights), the adapter also yields instruction-following in Gemma, showing great generalization.
 - **New Evaluations.** We hear the reviewers’ concerns about our AlpacaEval-based evaluations. We believe them to be valid, especially since none of our methods, e.g., lead to very long responses (in fact, the base models’ responses are almost always longer.) However, we have added IFEval and MTBench evaluations. Overall, models perform relatively poorly on IFEval relative to instruction tuning (showing that they are not as good at capturing details), and MTBench gives qualitatively similar results to AlpacaEval for Llama. For Olmo and Gemma, only GSM outperforms the base model on MTBench, but for Olmo in particular, looking at the samples (in our appendix,) we see the AlpacaEval evaluations as more indicative of the relative quality of response to the base models.
 - **Additional Instruction-Tuning Dataset** We hear reviewers’ concerns about using LIMA as our instruction-tuning dataset and ran additional experiments using Tulu-v2 data, finding similar conclusions to the LIMA experiments.

Overall, we believe these new experiments complement the existing ablations and evaluations we submitted in the initial draft, strengthening the paper even further.

---

### Public Comment · ~Pinzhen_Chen1 · 2024-11-24
**Question: difference between pre-training and response tuning?**

Thanks for your work! After reading the paper, I am curious about the difference between response tuning (Sec 4.1) and the LLM pre-training stage. In terms of the training objective, they seem equivalent. If so, I then wonder why the former can lead to better instruction following but not the latter. Is it mostly due to the style/domain/format of the (response) data? A paradox here is that an arbitrary (pre-training) text can be a suitable response for some instructions, making LLM pre-training itself "response tuning".

---

> ### Author Response · Authors · 2024-11-25
> **It's mainly about the distribution**
>
> Hi Pinzhen,
>
> Thanks for your question. So, the key difference is in what LIMA responses are like--structured as responses to some relatively concise question--vs. what pretraining documents are like. Yes, any text (including pretraining text) is a suitable response for some instruction; e.g., for any text $T$, I could say "Generate the string $T$". One could say that it's the style/format of responses, and we think that's a lot of it, but it's not quite so obvious to us it's all that's going on---there's also a notion of the responses being good responses _for some instruction in some reasonable distribution of instructions people tend to ask_. I'd guess that most pretraining texts are not good responses for the kinds of questions that are evaluated in modern LLM evaluations. Responses in instruction-tuning datasets are generally self-contained given the dialogue so far, whereas a pretraining document might not make a ton of sense as a response due to a lack of context, e.g., people might not tend not to ask "Give me the twenty-third through twenty-sixth page of the unix manual for grep," but that might be a pretraining document.
>
> As another simple effect, response tuning also teaches the model not to generate the instruction-response delimiter tokens, like `<|assistant|>` or `<|user|>`; I mention this because it seems important in our rule-based system to explicitly disallow tokens that would begin the generation of these strings. Base models see the formatting tags and really tend to like to repeat them; response tuning (and single-task finetuning for that matter) teach the model not to repeat these tokens.
>
> So, response tuning is certainly maximizing the likelihood of a bunch of text, as is pretraining -- response-tuning differs in that (1) the distribution of responses in instruction-tuning datasets is different from the distribution of pretraining data, and (2) (though you could consider this part of (1)) you condition on these formatting tags and learn not to generate them when you see them.

---

> > ### Public Comment · ~Pinzhen_Chen1 · 2024-11-28
> >
> > Thank you for providing a detailed explanation and sharing your insights!

---

### Meta-Review · Area_Chair_spLm · 2024-12-19

**Metareview:**

The paper notes that you can achieve traditional instruction tuning effects by just tuning on responses and then provides a 3 rule method to induce instruction following abilities without actually instruction tuning. The effects of how instruction tuning abilities happen and their interplay with pretraining dynamics are of high interest to the entire LLM community at large and actionable insights into this topic would be high impact.

The paper claims overall to make generalizable observations about instruction vs response tuning on base models ("Our goal is insight through ablations") but unfortunately I agree with the reviewers who think that the results from the models provided show no clear insights. Normally I wouldn't be too concerned with the choice of models but in this case they are very relevant as noted by reviewer uWBN.  Olmo is the only model they can guarantee did not have instruction tuned data / was midtrained but nothing is mentioned to be best of my knowledge in the Olmo paper about how they filtered for instruction data in the pretraining corpus. This places their fundamental claim on shaky grounds. The three rules of the rule based approach are also seemingly arbitrary and not fully justified with respect to the claims originally made. The primary eval, AlpacaEval, is also not particularly reliable as noted by reviewer aAKp - being entirely GPT based (but I do appreciate the examples shown which qualitatively show the differences between the various models).

The paper has a lot of potential and I believe with some (significant) revisions it will have high impact but it is not suitable for publication in its current form.

**Additional Comments On Reviewer Discussion:**

I have to discount the score of 10 by reviewer eN6V as the text of the review doesn't match up with what a score of 10 should be. They themselves in the weaknesses section point out that all the confounding variables must be clearly separated and many of the other reviewers agree that this work does not. Of the remaining 2/3 recommend rejection though the authors provide a good faith attempt to answer all the reviewer's questions during the rebuttal phase.

---

### Decision · Program_Chairs · 2025-01-22

Reject